# SGLT2 inhibition modulates NLRP3 inflammasome activity via ketones and insulin in diabetes with cardiovascular disease

So Ra Kim [1,2,3,14], Sang-Guk Lee [4,14], Soo Hyun Kim[1], Jin Hee Kim [1,5], Eunhye Choi[4], Wonhee Cho[6], John Hoon Rim[5,7], Inhwa Hwang[5,8], Chan Joo Lee[9], Minyoung Lee[1], Chang-Myung Oh[10], Justin Y. Jeon[6], Heon Yung Gee [5,7], Jeong-Ho Kim [4], Byung-Wan Lee [1,2,11], Eun Seok Kang[1,2,5,11], Bong-Soo Cha[1,2,5,11], Myung-Shik Lee [1,12], Je-Wook Yu[5,8], Jin Won Cho[13], Jung-Sun Kim [9]✉ & Yong-ho Lee [1,2,5,11,13]✉

Sodium–glucose cotransporter 2 (SGLT2) inhibitors reduce cardiovascular events in humans with type 2 diabetes (T2D); however, the underlying mechanism remains unclear. Activation of the NLR family, pyrin domain-containing 3 (NLRP3) inflammasome and subsequent interleukin (IL)-1β release induces atherosclerosis and heart failure. Here we show the effect of SGLT2 inhibitor empagliflozin on NLRP3 inflammasome activity. Patients with T2D and high cardiovascular risk receive SGLT2 inhibitor or sulfonylurea for 30 days, with NLRP3 inflammasome activation analyzed in macrophages. While the SGLT2 inhibitor's glucose-lowering capacity is similar to sulfonylurea, it shows a greater reduction in IL-1β secretion compared to sulfonylurea accompanied by increased serum β-hydroxybutyrate (BHB) and decreased serum insulin. Ex vivo experiments with macrophages verify the inhibitory effects of high BHB and low insulin levels on NLRP3 inflammasome activation. In conclusion, SGLT2 inhibitor attenuates NLRP3 inflammasome activation, which might help to explain its cardioprotective effects.

[1] Division of Endocrinology and Metabolism, Department of Internal Medicine, Yonsei University College of Medicine, 50-1 Yonsei-ro, Seodaemun-gu, Seoul 03722, Republic of Korea. [2] Graduate School, Yonsei University College of Medicine, 50-1 Yonsei-ro, Seodaemun-gu, Seoul 03722, Republic of Korea. [3] Department of Hospital Medicine, Yongin Severance Hospital, Yonsei University College of Medicine, Yongin 16995, Republic of Korea. [4] Department of Laboratory Medicine, Yonsei University College of Medicine, 50-1 Yonsei-ro, Seodaemun-gu, Seoul 03722, Republic of Korea. [5] Brain Korea 21 PLUS Project for Medical Science, Yonsei University College of Medicine, 50-1 Yonsei-ro, Seodaemun-gu, Seoul 03722, Republic of Korea. [6] Exercise Medicine Center for Diabetes and Cancer Patients, ICONS, Yonsei University, Seoul 03722, Republic of Korea. [7] Department of Medicine, Physician-Scientist Program, Yonsei University Graduate School of Medicine, Seoul 03722, Republic of Korea. [8] Department of Microbiology and Immunology, Institute for Immunology and Immunological Diseases, Yonsei University College of Medicine, 50-1 Yonsei-ro, Seodaemun-gu, Seoul 03722, Republic of Korea. [9] Cardiology Division, Severance Cardiovascular Hospital, Yonsei University College of Medicine, 50-1 Yonsei-ro, Seodaemun-gu, Seoul 03722, Republic of Korea. [10] Department of Biomedical Science and Engineering, Gwangju Institute of Science and Technology, Gwangju 61005, Republic of Korea. [11] Institute of Endocrine Research, Yonsei University College of Medicine, 50-1 Yonsei-ro, Seodaemun-gu, Seoul 03722, Republic of Korea. [12] Severance Biomedical Science Institute, Yonsei Biomedical Research Institute, Yonsei University College of Medicine, 50-1 Yonsei-ro, Seodaemun-gu, Seoul 03722, Republic of Korea. [13] Department of Systems Biology, Glycosylation Network Research Center, Yonsei University, Seoul 03722, Republic of Korea. [14]These authors contributed equally: So Ra Kim, Sang-Guk Lee. ✉email: KJS1218@yuhs.ac; yholee@yuhs.ac

Cardiovascular disease (CVD) is a major cause of morbidity and mortality in patients with type 2 diabetes (T2D), with a two- to fourfold increase in incidence compared with patients without diabetes[1]. Diabetes produces an abnormal metabolic state, including chronic hyperglycemia, dyslipidemia, insulin resistance, and inflammation, which can lead to the development of atherosclerosis, a pathologic process implicated in symptomatic CVD events. A large, multiprotein complex oligomerized in the cytoplasm of innate immune cells, known as the NLR family, pyrin domain-containing 3 (NLRP3) inflammasome, is stimulated to secrete pathogenic inflammatory cytokines, specifically interleukin-1β (IL-1β)[2]. This is involved in the molecular etiology of numerous chronic inflammatory diseases, including diabetes, non-alcoholic steatohepatitis, gout, atherosclerosis, and heart failure[2–7].

Sodium–glucose cotransporter 2 (SGLT2) inhibitors lower serum glucose by increasing urinary glucose excretion[8]. The recently published EMPA-REG OUTCOME study demonstrated that, in patients with T2D and high CVD risk, empagliflozin reduced adverse cardiac events by 14%, which resulted in a 38% reduction in cardiovascular (CV) mortality[9]. It is the first antidiabetic agent that reduced CV events beyond glycemic control. Subsequently, the CANVAS Program also achieved comparable effects with another SGLT2 inhibitor[10]. The unique action of SGLT2 inhibitors corrects several metabolic and hemodynamic abnormalities that are risk factors for CVD, by decreasing serum glucose, body weight, and blood pressure and by increasing diuresis[8]. However, other underlying mechanisms to explain the cardioprotective effects of SGLT2 inhibitors are as yet unclear.

Based on their pharmacologic profile, SGLT2 inhibitors cause mild increases in serum ketone bodies, i.e., β-hydroxybutyrate (BHB)[8]. BHB is a convenient carrier of energy from adipocytes to peripheral tissues, particularly in brain, heart, and skeletal muscles during prolonged fasting or exercise[11]. Beyond acting as a metabolite, BHB has important cellular signaling roles. Youm et al. revealed that BHB suppresses activation of the NLRP3 inflammasome and reduces IL-1β production in macrophages and mice[12]. However, there is no report of whether increased serum BHB concentration inhibits NLRP3 inflammasome activation in humans. A recent study demonstrated that fasting and refeeding can differentially regulate NLRP3 inflammasome activation in human peripheral blood mononuclear cells (PBMCs)[13]. This indicates that the dynamic regulation of NLRP3 inflammasome activation can be assessed using human PBMCs.

On the basis of the pathogenic effect of the NLRP3 inflammasome on CVD and the therapeutic role of BHB on its inhibition, the present study demonstrates that the SGLT2 inhibitor blocks NLRP3 inflammasome activation by raising circulating levels of BHB in patients with T2D at high CV risk, which could explain their cardioprotective effects. To exclude glucose-lowering effects on inhibition of NLRP3 inflammasome activation, we used an active comparator, namely sulfonylurea instead of placebo.

## Results

**Baseline characteristics of study participants.** Among 88 screened participants, 71 met the inclusion criteria and were randomly assigned to treatment with empagliflozin (SGLT2 inhibitor) or glimepiride (sulfonylurea). Ten participants withdrew prematurely from the trial, which led to 61 participants completing the study (empagliflozin, 29; and glimepiride, 32) (Fig. 1).

Baseline characteristics of the study participants are summarized in Table 1. Participants were matched for demographic and biochemical variables. As metformin is first-line therapy, most participants in both groups were taking it and continued on the

same dosage during the study period. Adverse events related to study drugs were uncommon, although discontinuation of study drug occurred once in each group (Supplementary Table 1).

**Effects of SGLT2 inhibitor on metabolic parameters.** Despite a similar glucose-lowering effect in the two groups (Table 2 and Fig. 2a), distinct patterns of change in metabolic parameters were observed in the SGLT2 inhibitor group. SGLT2 inhibitor caused a significant increase in fasting serum BHB, ~3.9-fold from baseline (Fig. 2b) and a significant decrease in serum uric acid and fasting serum insulin (Fig. 2c, d, respectively) accompanied by an increase in fasting serum free fatty acid (FFA) (Fig. 2e), while sulfonylurea had no significant effects on these measurements. SGLT2 inhibitor induced significant improvement in insulin sensitivity (Fig. 2f, g), while sulfonylurea led to increased insulin secretion (Fig. 2h). SGLT2 inhibitor significantly decreased body weight, with a mean change of −2.5% (Fig. 2i).

**SGLT2 inhibitor suppresses NLRP3 inflammasome activation.** Regarding the primary endpoint of the current study, IL-1β secretion levels were measured from baseline to end of treatment in isolated macrophages after exposure to 2 mM ATP or 0.2 mM palmitate as an NLRP3 trigger, with 0.1 μg/mL LPS priming. In response to ATP stimulation, IL-1β secretion was significantly reduced after both SGLT2 inhibitor and sulfonylurea treatment (3733 ± 360 to 2549 ± 320 pg/mL, $P < 0.001$; and 3777 ± 485 to 3121 ± 345 pg/mL, $P = 0.01$, respectively) (Fig. 3a). However, SGLT2 inhibitor showed a greater reduction in IL-1β secretion compared with sulfonylurea (time × group interaction $P = 0.002$), which remained significant after adjustment for body weight change (time × group interaction $P = 0.02$). This tendency was potentiated under the condition of inflammasome stimulation by palmitate (time × group interaction $P < 0.001$, Fig. 3b). To further evaluate metabolic factors associated with changes in inflammasome activity, correlation analyses were conducted. Changes in fasting serum insulin or BHB levels were significantly correlated with changes in NLRP3 inflammasome activity (Supplementary Fig. 1).

In parallel with the IL-1β effect, tumor necrosis factor-α (TNF-α) secretion was significantly reduced after SGLT2 inhibitor treatment in response to ATP and palmitate stimulation (262 ± 61 to 145 ± 22 pg/mL, $P < 0.001$; and 271 ± 54 to 148 ± 26 pg/mL, $P < 0.001$, respectively) (Fig. 3c, d). Alternatively, sulfonylurea treatment had no effect on TNF-α secretion in response to ATP and palmitate stimulation (210 ± 32 to 204 ± 30 pg/mL, $P = 0.41$; and 206 ± 35 to 215 ± 31 pg/mL, $P = 0.64$, respectively). A previous study found that TNF-α may cause atherosclerosis through pro-inflammatory actions on leukocytes, endothelial cells, and adipocytes[14]. A number of studies have described the protective effects of TNF antagonists on vascular diseases[15]. To rule out the effect of body weight loss on changes in inflammasome activity, correlation analyses were conducted. As a result, there was no significant correlation between changes in body weight and changes in IL-1β or TNF-α release (Supplementary Fig. 2).

Next, we measured the mRNA levels of *IL1B*, *TNFA*, and *NLRP3* in unstimulated macrophages following 30-day treatment with sulfonylurea or SGLT2 inhibitor. The transcripts encoding IL-1β were significantly decreased following SGLT2 inhibitor treatment (Fig. 3e). Transcript levels of *TNFA* and *NLRP3* tended to decrease after SGLT2 inhibitor treatment, but not statistically significant (Fig. 3f, g). SGLT2 inhibitor significantly decreased the LPS- and ATP-induced processing of the biologically active form of IL-1β in cell lysates (Fig. 3h). RNA sequencing with gene ontology (GO) enrichment analysis identified two clusters of

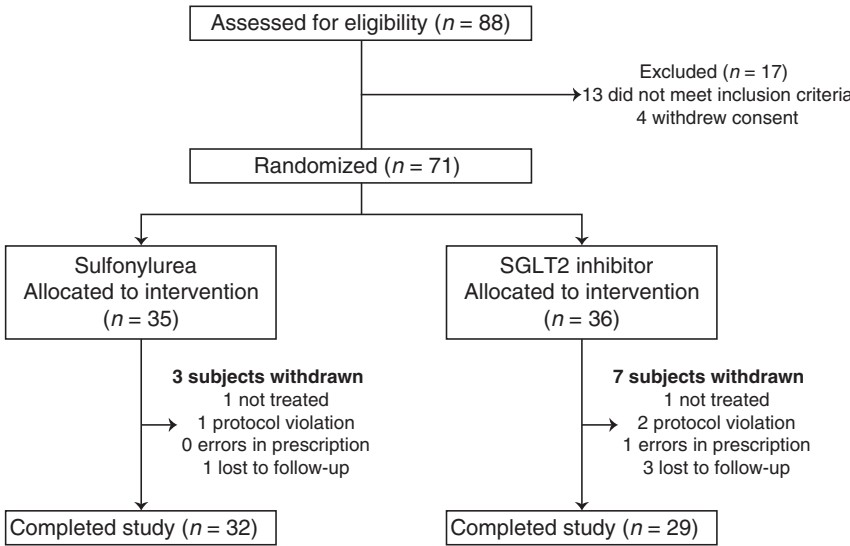

**Fig. 1 Participant flow diagram.** Numbers of participants who were initially screened, underwent random allocation, withdrew, and were included in the final analysis. SGLT2 sodium–glucose cotransporter 2.

upregulated and downregulated genes in SGLT2 inhibitor group compared with sulfonylurea group. The downregulated genes were highly enriched for immune-related receptor signaling pathway, activation of immune response, and regulation of immune system process, whereas the upregulated cluster was enriched for genes involved in cellular components such as vesicles and extracellular region (Supplementary Fig. 3).

Regarding serum IL-1β and IL-18 levels, despite a tendency to decrease after SGLT2 inhibitor treatment, there were no significant changes in those concentrations (Supplementary Table 2). This may be attributable to the following factors: First, serum IL-1β levels are often undetectable even in diseases with clear evidence of increased IL-1 activity[5,16–18]. IL-1β exerts its effect in an autocrine/paracrine fashion and, consequently, the detected levels of IL-1β in the serum may be very low[19]; in our study, 37 out of 61 participants had serum IL-1β levels below the detection limits of the assay used. Second, extensive use of aspirin or statin in both sulfonylurea and SGLT2 inhibitor groups might attenuate changes in serum pro-inflammatory cytokines[20]. Finally, as activation of the NLRP3 inflammasome requires two independent steps: priming and triggering[21], IL-1β signaling in the immune cells of vessel tissues or of atherosclerotic lesions enriched for various endogenous danger signals, rather than circulating IL-1β levels, might be a major determinant of atheroma formation or heart failure[5,22,23]. In the present study, we identified that treatment with an SGLT2 inhibitor can reduce the activation of NLRP3 inflammasome in human macrophages after stimulation with LPS and ATP/palmitate.

**Effects of BHB, glucose, and insulin on NLRP3 inflammasome.** Although it has been reported that BHB blocks activation of the NLRP3 inflammasome-IL-1β process by preventing K$^+$ efflux and reducing apoptosis-associated speck-like protein containing a caspase recruitment domain (ASC) oligomerization and speck formation[12], we found that its effective concentration was rather high compared with the modest increase seen after SGLT2 inhibitor treatment. Therefore, other mediators besides BHB may play important roles in modulating NLRP3 inflammasome activity in humans. As SGLT2 inhibitor significantly reduced fasting serum levels of insulin as well as glucose, we performed ex vivo experiments using human macrophages to investigate whether these metabolites could affect NLRP3 inflammasome activity.

We confirmed that BHB dose-dependently inhibited IL-1β secretion from human macrophages stimulated by LPS and ATP (Fig. 4a). Co-treatment with 2-deoxyglucose (2-DG), a non-metabolizable glucose analog that blocks glycolysis and has been used to mimic a condition of glucose starvation[24], significantly decreased IL-1β secretion in a dose-dependent manner (Fig. 4b). Inversely, a high glucose condition (25 mM) markedly increased IL-1β production, which was blocked by administration of 2-DG (Fig. 4c). Regarding the role of insulin on NLRP3 inflammasome, 10 nM insulin significantly elevated IL-1β secretion (Fig. 4d). Furthermore, co-treatment with insulin attenuated the inhibitory effect of BHB on NLRP3 inflammasome activation (Fig. 4e).

## Discussion

We demonstrate that an SGLT2 inhibitor significantly suppresses NLRP3 inflammasome activation and subsequent secretion of IL-1β in human macrophages, via increased serum BHB levels and decreased serum levels of insulin, among patients with T2D and CVD, regardless of glycemic control. Furthermore, ex vivo experiments with human macrophages verify the mediatory roles of BHB, insulin, and glucose on the modulating activity of the NLRP3 inflammasome.

NLRP3 inflammasome is involved in the molecular etiology of atherosclerosis and heart failure[2,4,5,7]. Mice with constitutively increased IL-1 signaling because of IL-1 receptor antagonist deficiency develop a transmural arterial inflammation leading to lethal aneurysms[25]. In human atherosclerotic plaques, the expression of IL-1β appears to correlate with the progression of atherosclerotic plaques[26]. In an in vitro study, IL-1β and TNF-α decreased collagen synthesis and activated collagen degradation in rat cardiac fibroblasts, suggesting that IL-1β and TNF-α may contribute to ventricular dilation and heart failure[3].

Although there is a paucity of large clinical trial data, a number of observational, pilot, and preclinical studies suggest a beneficial role of IL-1 blockade in the pathological processes of CVD. Experimental studies in atherosclerosis-prone animals have consistently shown that genetic deletion or pharmacological inhibition of IL-1 signaling reduces the formation and progression of atherosclerotic plaques[5,7]. For example, transplantation of bone marrow cells from *NLRP3*- or *IL1*-deficient mice into mice models of atherosclerosis resulted in significant amelioration of aortic atherogenesis[4]. In terms of heart failure, when exposed to

**Table 1 Baseline characteristics of study participants.**

| Baseline characteristics | Sulfonylurea (n = 32) | SGLT2 inhibitor (n = 29) | P values |
|---|---|---|---|
| Demographics | | | |
| Age (years) | 64.9 ± 8.60 | 63.9 ± 9.18 | 0.64 |
| Male sex [n (%)] | 22 (68.8) | 24 (82.8) | 0.20 |
| Body weight (kg) | 69.5 ± 11.0 | 73.0 ± 14.8 | 0.30 |
| BMI (kg/m$^2$) | 26.0 (24.3–27.9) | 26.3 (24.3–28.0) | 0.98 |
| Currently smoking [n (%)] | 3 (9.38) | 7 (24.1) | 0.17 |
| Systolic blood pressure (mmHg) | 129.3 ± 14.0 | 127.8 ± 15.4 | 0.71 |
| Diastolic blood pressure (mmHg) | 73.6 ± 10.1 | 76.3 ± 11.3 | 0.33 |
| Duration of diabetes (years) | 7.58 (3.39–14.1) | 7.58 (4.00–12.6) | 0.83 |
| Cardiovascular disease | | | |
| History of ACS[a] [n (%)] | 13 (40.6) | 16 (55.2) | 0.26 |
| History of AMI [n (%)] | 9 (28.1) | 8 (27.6) | 0.96 |
| Multi-vessel CAD [n (%)] | 22 (68.8) | 23 (79.3) | 0.35 |
| PTCA [n (%)] | 23 (71.9) | 18 (62.1) | 0.42 |
| Coronary artery bypass graft [n (%)] | 3 (9.38) | 3 (10.3) | >0.99 |
| Gluco-metabolic indices | | | |
| HbA$_{1C}$ (%) | 7.25 (6.75–8.00) | 6.90 (6.45–7.80) | 0.25 |
| Glycated albumin (%) | 17.4 (15.7–21.0) | 17.2 (15.3–19.9) | 0.50 |
| Fasting serum glucose (mg/dL) | 139.0 (127.8–171.5) | 128.0 (123.5–147.0) | 0.05 |
| Fasting serum BHB (mM) | 0.07 ± 0.07 | 0.06 ± 0.04 | 0.21 |
| Uric acid (mg/dL) | 4.96 ± 1.43 | 4.75 ± 1.19 | 0.54 |
| AST (IU/L) | 24.0 (19.3–33.0) | 22.0 (19.5–29.0) | 0.41 |
| ALT (IU/L) | 24.0 (19.0–34.0) | 27.0 (18.5–35.0) | 0.60 |
| Total cholesterol (mg/dL) | 139.0 (118.5–158.0) | 133.0 (117.5–147.0) | 0.30 |
| Triglyceride (mg/dL) | 126.5 (101.3–183.5) | 143.0 (104.0–188.0) | 0.76 |
| HDL cholesterol (mg/dL) | 43.6 ± 10.2 | 42.0 ± 9.77 | 0.54 |
| LDL cholesterol (mg/dL) | 65.2 (43.2–75.2) | 63.4 (47.4–71.2) | 0.40 |
| Creatinine (mg/dL) | 0.85 (0.73–1.00) | 0.84 (0.77–0.92) | 0.61 |
| eGFR CKD-EPI (mL/min per 1.73 m$^2$) | 86.0 (74.8–94.5) | 92.0 (83.0–98.5) | 0.08 |
| Insulin secretory/resistant indices | | | |
| Fasting serum insulin (μU/mL) | 9.60 (7.36–16.8) | 8.09 (5.31–11.9) | 0.08 |
| Fasting serum FFA (μEq/L) | 426.5 (351.5–527.0) | 412.0 (284.0–517.0) | 0.41 |
| HOMA-IR | 3.89 (2.32–7.42) | 2.72 (1.74–4.40) | 0.05 |
| QUICKI | 0.32 ± 0.03 | 0.33 ± 0.03 | 0.05 |
| HOMA-β (%) | 46.2 (29.8–92.4) | 48.7 (27.6–55.5) | 0.42 |
| Drug use | | | |
| Antiplatelet/anticoagulant agents [n (%)] | 30 (93.8) | 27 (93.1) | >0.99 |
| Statin [n (%)] | 29 (90.6) | 27 (93.1) | >0.99 |
| Fibrate [n (%)] | 3 (9.38) | 4 (13.8) | 0.70 |
| ACE inhibitor/ARB [n (%)] | 20 (62.5) | 18 (62.1) | 0.97 |
| Diuretics [n (%)] | 4 (12.5) | 2 (6.90) | 0.67 |
| Calcium channel blockers [n (%)] | 6 (18.8) | 4 (13.8) | 0.74 |
| Beta blockers [n (%)] | 22 (68.8) | 15 (51.7) | 0.17 |
| Metformin [n (%)] | 31 (96.9) | 26 (89.7) | 0.34 |

Mann–Whitney U or two-sample Student's t test for continuous variables and a Pearson $\chi^2$ test for categorical variables; continuous variables are described as mean ± SD for parametric variables and as median (interquartile range) for non-parametric variables. Source data are provided as a Source Data file.
*ACE* angiotensin-converting enzyme, *ACS* acute coronary syndrome, *ALT* alanine aminotransferase, *AMI* acute myocardial infarction, *ARB* angiotensin II receptor blocker, *AST* aspartate aminotransferase, *BHB* β-hydroxybutyrate, *BMI* body mass index, *CAD* coronary artery disease, *CKD-EPI* Chronic Kidney Disease Epidemiology Collaboration, *eGFR* estimated glomerular filtration rate, *FFA* free fatty acid, *HbA$_{1C}$* glycated hemoglobin, *HDL* high-density lipoprotein, *HOMA-β* homeostatic model assessment of pancreatic β-cell function, *HOMA-IR* homeostatic model assessment of insulin resistance, *LDL* low-density lipoprotein, *n* number of patients, *PTCA* percutaneous transluminal coronary angioplasty, *QUICKI* quantitative insulin sensitivity check index, *SD* standard deviation, *SGLT2* sodium–glucose cotransporter 2.
[a]History of AMI or unstable angina.

ischemia-reperfusion injury, mice that lack the *NLRP3* gene have a smaller infarct size and better cardiac function than wild-type mice[27]. In the recent CANTOS (Canakinumab Anti-inflammatory Thrombosis Outcomes Study) trial, canakinumab, a monoclonal antibody against IL-1β, significantly prevented recurrent CV events among patients with a prior myocardial infarction (MI) and high levels of pro-inflammatory biomarkers[28]. This randomized controlled trial (RCT) implicates that reducing low-grade inflammation in CVD patients without affecting blood cholesterol levels can decrease the risk of developing CV events. In addition, it confirms the clinical importance of the proatherogenic characteristics of IL-1β in humans.

Previously, subgroup analyses from EMPA-REG OUT-COME or CANVAS Program did not show a significant reduction in MI or stroke by SGLT2 inhibitors[9,10]; however, these secondary endpoints could not be proven with relatively small numbers of trial participants. In a recent meta-analysis of 71 RCTs[29] and the large multinational CVD-REAL 2 study[30], SGLT2 inhibitors significantly decreased MI and stroke events, as well as all-cause death and hospitalization for heart failure, suggesting their anti-atherothrombotic effects. Our study suggests that SGLT2 inhibitors offer not only improvements in metabolic profiles but also inhibition of IL-1β secretion and other pro-inflammatory cytokines, potentially

**Table 2 Effects of sulfonylurea and SGLT2 inhibitor on metabolic parameters.**

| | Sulfonylurea ($n = 32$) | | | SGLT2 inhibitor ($n = 29$) | | | $P_{time \times group}$ |
|---|---|---|---|---|---|---|---|
| | Baseline | Day 30 | P | Baseline | Day 30 | P | |
| Body weight (kg) | 69.5 ± 11.0 | 69.1 ± 10.3 | 0.47 | **73.0 ± 14.8** | **71.2 ± 14.1** | **<0.001** | **<0.001** |
| Glycated albumin (%) | **17.4 (15.7–21.0)** | **16.9 (14.8–18.9)** | **<0.001** | **17.2 (15.3–19.9)** | **16.3 (14.4–18.3)** | **0.01** | 0.32[†] |
| Fasting serum glucose (mg/dL) | 139.0 (127.8–171.5) | 131.0 (112.0–165.8) | 0.14 | 128.0 (123.5–147.0) | 124.0 (112.5–138.0) | 0.13 | 0.64[†] |
| Fasting serum BHB (mM) | 0.07 ± 0.07 | 0.07 ± 0.07 | 0.60 | **0.06 ± 0.04** | **0.20 ± 0.19** | **<0.001** | **<0.001** |
| Fasting serum BHB (fold increase) | 1 | 1.29 | | 1 | 3.91 | | |
| Uric acid (mg/dL) | 4.96 ± 1.43 | 5.00 ± 1.34 | 0.94 | **4.75 ± 1.19** | **4.39 ± 1.08** | **0.01** | 0.08 |
| AST (IU/L) | 24.0 (19.3–33.0) | 25.0 (20.5–31.5) | 0.95 | 22.0 (19.5–29.0) | 23.0 (19.0–27.0) | 0.54 | 0.45[†] |
| ALT (IU/L) | 24.0 (19.0–34.0) | 27.5 (20.3–36.0) | 0.94 | 27.0 (18.5–35.0) | 24.0 (17.0–28.5) | 0.07 | 0.06[†] |
| Total cholesterol (mg/dL) | 139.0 (118.5–158.0) | 137.5 (128.0–147.8) | 0.48 | 133.0 (117.5–147.0) | 129.0 (115.0–146.0) | 0.86 | 0.61[†] |
| Triglyceride (mg/dL) | 126.5 (101.3–183.5) | 142.5 (99.3–182.3) | 0.96 | 143.0 (104.0–188.0) | 113.0 (87.5–180.5) | 0.15 | 0.18[†] |
| HDL cholesterol (mg/dL) | 43.6 ± 10.2 | 42.6 ± 8.46 | 0.36 | 42.0 ± 9.77 | 43.7 ± 10.7 | 0.27 | 0.15 |
| LDL cholesterol (mg/dL) | 65.2 (43.2–75.2) | 67.2 (53.5–82.0) | 0.11 | 63.4 (47.4–71.2) | 57.2 (47.6–73.3) | 0.96 | 0.76[†] |
| Creatinine (mg/dL) | 0.85 (0.73–1.00) | 0.86 (0.73–1.00) | 0.41 | 0.84 (0.77–0.92) | 0.84 (0.77–0.95) | 0.14 | 0.54[†] |
| eGFR CKD-EPI (mL/min per 1.73 m²) | 86.0 (74.8–94.5) | 85.5 (74.3–93.5) | 0.39 | 92.0 (83.0–98.5) | 91.0 (78.5–98.0) | 0.16 | 0.59[†] |
| Fasting serum insulin (μU/mL) | 9.60 (7.36–16.8) | 10.7 (6.15–19.6) | 0.34 | **8.09 (5.31–11.9)** | **6.09 (4.50–8.94)** | **<0.001** | **0.002[†]** |
| Fasting serum FFA (μEq/L) | 426.5 (351.5–527.0) | 416.0 (333.5–536.8) | 0.60 | **412.0 (284.0–517.0)** | **523.0 (417.5–637.5)** | **<0.001** | **0.01[†]** |
| HOMA-IR | 3.89 (2.32–7.42) | 3.46 (2.06–7.05) | 0.98 | **2.72 (1.74–4.40)** | **1.92 (1.32–2.91)** | **<0.001** | **0.01[†]** |
| QUICKI | 0.32 ± 0.03 | 0.32 ± 0.03 | 0.85 | **0.33 ± 0.03** | **0.35 ± 0.03** | **<0.001** | **0.01** |
| HOMA-β (%) | **46.2 (29.8–92.4)** | **70.7 (25.8–112.3)** | **0.02** | 48.7 (27.6–55.5) | 38.6 (26.5–52.4) | 0.09 | **0.004[†]** |
| Spot urine UGCR (mg/mg)* | 0.07 (0.05–0.26) | 0.07 (0.05–0.16) | 0.22 | **0.11 (0.06–0.38)** | **43.3 (33.1–55.1)** | **<0.001** | **<0.001[†]** |

Bold values indicate statistical significance. *Urinary glucose levels are expressed as UGCR to minimize the influence of variations of kidney function using the following formula: spot urine UGCR (mg/mg) = [spot urine glucose (mg/dL) / spot urine creatinine (mg/dL)]. Statistical significance for the time × group interaction was evaluated by using repeat-measures analysis of variance (ANOVA) ([†]Non-normally distributed variables were log transformed for analysis and back transformed for presentation). Two-sided paired $t$ test or Wilcoxon signed rank test; continuous variables are described as mean ± SD for parametric variables and as median (interquartile range) for non-parametric variables. Source data are provided as a Source Data file.
ALT alanine aminotransferase, AST aspartate aminotransferase, BHB β-hydroxybutyrate, CKD-EPI Chronic Kidney Disease Epidemiology Collaboration, eGFR estimated glomerular filtration rate, FFA free fatty acid, HDL high-density lipoprotein, HOMA-IR homeostatic model assessment of insulin resistance, LDL low-density lipoprotein, QUICKI quantitative insulin sensitivity check index, SD standard deviation, SGLT2 sodium–glucose cotransporter 2, UGCR urinary glucose-to-creatinine ratio.

reducing the development of CVD in high risk patients with diabetes.

Glycosuria resulting from SGLT2 inhibition subtracts glucose from blood, and relative hypoinsulinemia reduces tissue glucose uptake, producing a compensatory increase in lipid oxidation and a concomitant rise in serum BHB levels[8]. In a previous in vivo study, the activity of NLRP3 inflammasome was suppressed in healthy individuals after 24-h fasting[13], which can significantly increase BHB levels, although this study did not measure serum levels of BHB in the fasted states. Along with reproducible outcomes of the inhibitory effect of BHB on NLRP3 inflammasome in our ex vivo study, we assumed that decreases in serum insulin level induced by SGLT2 inhibitor might also play a role in NLRP3 inflammasome inhibition. A recent study reported that insulin reinforced a pro-inflammatory pattern of macrophages via increased glucose uptake followed by subsequent glucose catabolism and production of reactive oxygen species, which eventually secreted IL-1β mediated by NLRP3 inflammasome activation[31]. Indeed, sulfonylurea increases insulin secretion, in turn reducing blood glucose levels. In contrast, SGLT2 inhibitor decreases blood glucose by renal glucose excretion, in turn decreasing serum insulin levels. These opposing changes in insulin levels might differentially regulate NLRP3 inflammasome in the two groups. In addition, similar to our findings, SGLT2 inhibitors have been shown to reduce serum uric acid levels through increasing renal clearance of uric acid[32], a potent activator of NLRP3 inflammasome[12]. Taken together, not only increased serum BHB, but also decreased serum levels of insulin, glucose, and uric acid, account for the overall inhibitory effect of SGLT2 inhibitor on the NLRP3 inflammasome.

A limitation of this study is that we did not assess whether the changes in inflammasome activity by SGLT2 inhibitor could be linked to the improvement in CV outcomes due to a relatively short-term trial design. However, both experimental evidence using NLRP3/IL1 knockout mice and the recent large-scale CANTOS trial indicate that therapeutic interference with IL-1β production or function can improve long-term CV outcomes[4,7,28]. The present study is a proof-of-concept RCT aimed to elucidate the glucose-independent mechanism of SGLT2 inhibitors

regarding CV protection. Preventive benefits were observed after 3 months of treatment in the EMPA-REG OUTCOME study; the present study demonstrates that only 1 month of SGLT2 inhibitor treatment can suppress inflammasome activation in individuals with T2D at high risk of CVD. Besides our findings, it is important to bear in mind that there may be multiplicity of other pathways or mechanisms which can be directly or indirectly involved in the protection of CVD by SGLT2 inhibitors. Nevertheless, the present study provides distinct evidence that SGLT2 inhibitor suppresses NLRP3 inflammasome activation in patients with T2D at high risk of CVD.

In conclusion, the present study shows that SGLT2 inhibitor treatment in patients with T2D at high risk of CVD attenuates NLRP3 inflammasome activation and secretion of IL-1β, which has a pathogenic effect on both T2D and CVD, in part via increased serum BHB and decreased serum insulin, glucose, and uric acid (Fig. 5). Moreover, the effects of BHB and insulin on NLRP3 inflammasome activation have also been verified ex vivo. The present data suggest that these mechanisms might help to explain the cardioprotective effects of SGLT2 inhibitor in humans.

## Methods

**Study design**. The present prospective, randomized, open, active-controlled, 2-arm parallel interventional study was carried out at Severance Hospital between November 2016 and July 2017. This RCT complied with the Declaration of Helsinki, and is registered at clinicaltrials.gov. (NCT02964572). The protocol was approved by the Institutional Review Board at Severance Hospital (4-2016-0795). All participants provided their written informed consent.

**Patients**. Eligible participants were aged 20–79 years with a diagnosis of T2D and high CV risk, having inadequate blood glucose control with metformin-based oral hypoglycemic agents (OHAs). High CV risk was defined as the presence of ≥1 of the following based on previous literatures[9,33]: (1) history of acute coronary syndrome (MI or unstable angina); (2) evidence of multi-vessel coronary artery disease documented by coronary angiography; (3) evidence of occlusive peripheral artery disease documented by angiography or ankle brachial index; (4) evidence of carotid atherosclerosis defined as presence of carotid artery plaques or ≥0.9 mm of peak carotid intima-media thickness documented by common carotid arterial ultrasound examinations[34]; (5) body mass index (BMI) ≥ 25 kg/m², with at least two of the following: history of hypertension, current smoking, or steatohepatitis. Inadequate blood glucose control was defined as: (1) glycated hemoglobin (HbA1C)

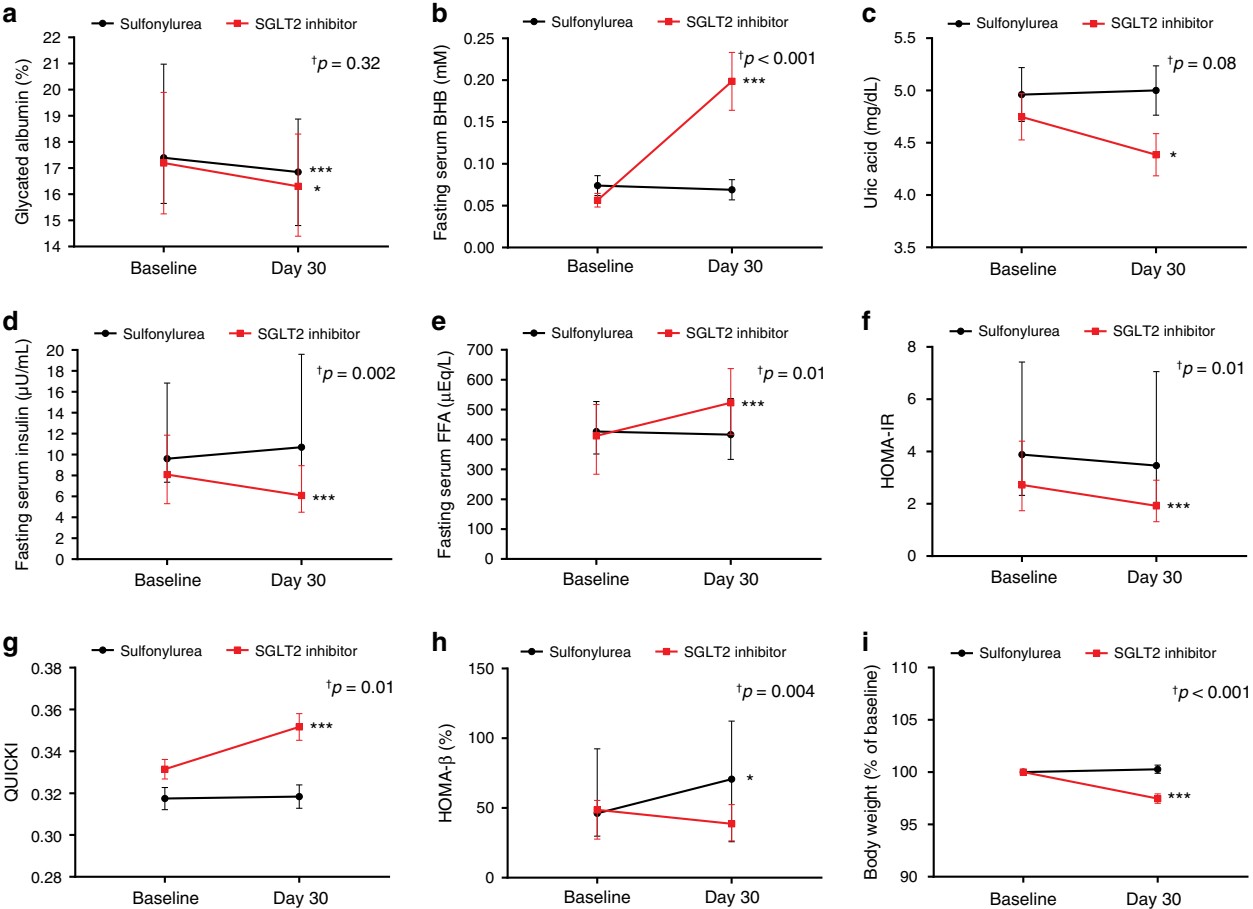

**Fig. 2 Effects of SGLT2 inhibitor and sulfonylurea on metabolic parameters. a–i** Changes in metabolic parameters from baseline to end of treatment (sulfonylurea group: $n = 32$, SGLT2 inhibitor group: $n = 29$). †Statistical significance for the time × group interaction evaluated by repeat-measures analysis of variance (ANOVA) (Non-normally distributed variables were log transformed for analysis and back transformed for presentation). Data are represented as mean ± SEM or median (interquartile range). Two-sided paired $t$ test or Wilcoxon signed rank test; *$P < 0.05$, **$P < 0.01$, and ***$P < 0.001$ versus baseline. BHB β-hydroxybutyrate, FFA free fatty acid, HOMA-IR homeostatic model assessment of insulin resistance, QUICKI quantitative insulin sensitivity check index, SEM standard error of the mean, SGLT2 sodium–glucose cotransporter 2. Source data are provided as a Source Data file.

≥6.5% checked within the last 3 months; (2) fasting serum glucose >120 mg/dL; or (3) random serum glucose >180 mg/dL. Key exclusion criteria included: (1) type 1 diabetes; (2) pregnant women; (3) estimated glomerular filtration rate ≤45 mL/min per 1.73 m$^2$; (4) active cancer; (5) any uncontrolled endocrine disorder; or (6) active infection. The inclusion criteria were amended from HbA$_1$c > 7% or fasting serum glucose ≥130 mg/dL to HbA$_1$c ≥ 6.5% or fasting serum glucose >120 mg/dL early in the course of the study.

**Randomization**. Participants who met the eligibility criteria were randomly assigned in a 1:1 ratio to receive empagliflozin or glimepiride once daily, stratifying for baseline HbA$_{1C}$ (<8% or ≥8%) and BMI (<25 kg/m$^2$ or ≥25 kg/m$^2$) using a computer-generated permuted block randomization (a block size of four). Each participant received 10 mg or 25 mg once-daily dose of empagliflozin, or glimepiride at an individualized dose according to risk of hypoglycemia and status of glycemic control (average dose was 2 mg). In participants who were previously treated with metformin-based OHAs, non-metformin OHAs were substituted with a study drug (either empagliflozin or glimepiride), or a study drug was added to OHAs. In drug naive participants, monotherapy with a study drug was initiated. OHA administration was maintained throughout the 30-day study period. A schematic illustrating the temporal organization of the study is shown in Supplementary Fig. 4.

**Outcome measures**. The primary endpoint was the group difference in the levels of IL-1β secreted from macrophages, before and after the administration of SGLT2 inhibitor or sulfonylurea. The secondary endpoints were the group differences in the levels of TNF-α secreted from macrophages, serum levels of BHB, insulin secretory/resistant indices, and other biochemical profiles (glycated albumin, glucose, uric acid, liver enzymes, lipid profiles, creatinine, and FFA), urinary glucose excretion, and body weight, before and after the administration of SGLT2 inhibitor or sulfonylurea. Quantitative PCR (qPCR) and immunoblot analyses for NLRP3, IL-1β, and TNF-α were performed and compared between

two treatment groups. For post-hoc analyses, RNA sequencing and GO enrichment analyses for differentially expressed genes between two groups were performed using PBMCs before and after treatment. In addition, changes in serum IL-1β and IL-18 levels from baseline to end of treatment were analyzed.

**Clinical and laboratory measurements**. A complete physical examination was conducted and current medications were recorded for all participants. For laboratory parameters, after overnight fasting, serum BHB was determined by an enzymatic assay using a commercial reagent from Nittobo Medical Co., Ltd (Tokyo, Japan) and the Hitachi 7600 analyzer. Serum glucose, glycated albumin, uric acid, FFA, lipid profiles, and creatinine were measured using a Hitachi 7600 automated chemistry analyzer (Hitachi High-Technologies Corporation, Tokyo, Japan). Fasting serum insulin was measured by electrochemiluminescence immunoassay using a Cobas e601 analyzer (Roche Diagnostics, GmbH, Germany). In addition, spot urine glucose and creatinine were measured on an AU680 chemistry system (Beckman Coulter, Inc., Brea, CA, USA) immediately prior to study initiation and at end of treatment. Baseline HbA$_{1C}$ was measured by immunoassay using an Integra 800 CTS (Roche Diagnostics). Pancreatic beta cell function and insulin sensitivity were assessed using the following indices[35]: homeostatic model assessment of pancreatic β-cell function (HOMA-β) = [(fasting serum insulin [μU/mL] × 20)/(fasting serum glucose [mmol/L] – 3.5)]; homeostatic model assessment of insulin resistance (HOMA-IR) = [(fasting serum insulin [μU/mL] × fasting serum glucose [mmol/L])/22.5]; and quantitative insulin sensitivity check index (QUICKI) = [1/(log(fasting serum glucose [mg/dL]) + log(fasting serum insulin [μU/mL]))][36]. The estimated glomerular filtration rate was derived from the Chronic Kidney Disease Epidemiology Collaboration (CKD-EPI)[37]. Serum IL-1β and IL-18 were measured with ELISA using human Quantikine HS ELISA kits (R&D Systems, MN, USA). The assay sensitivities (the minimum detectable levels) for IL-1β and IL-18 were 0.033 pg/mL and 1.25 pg/mL, respectively, determined by zero standard +2 standard deviation.

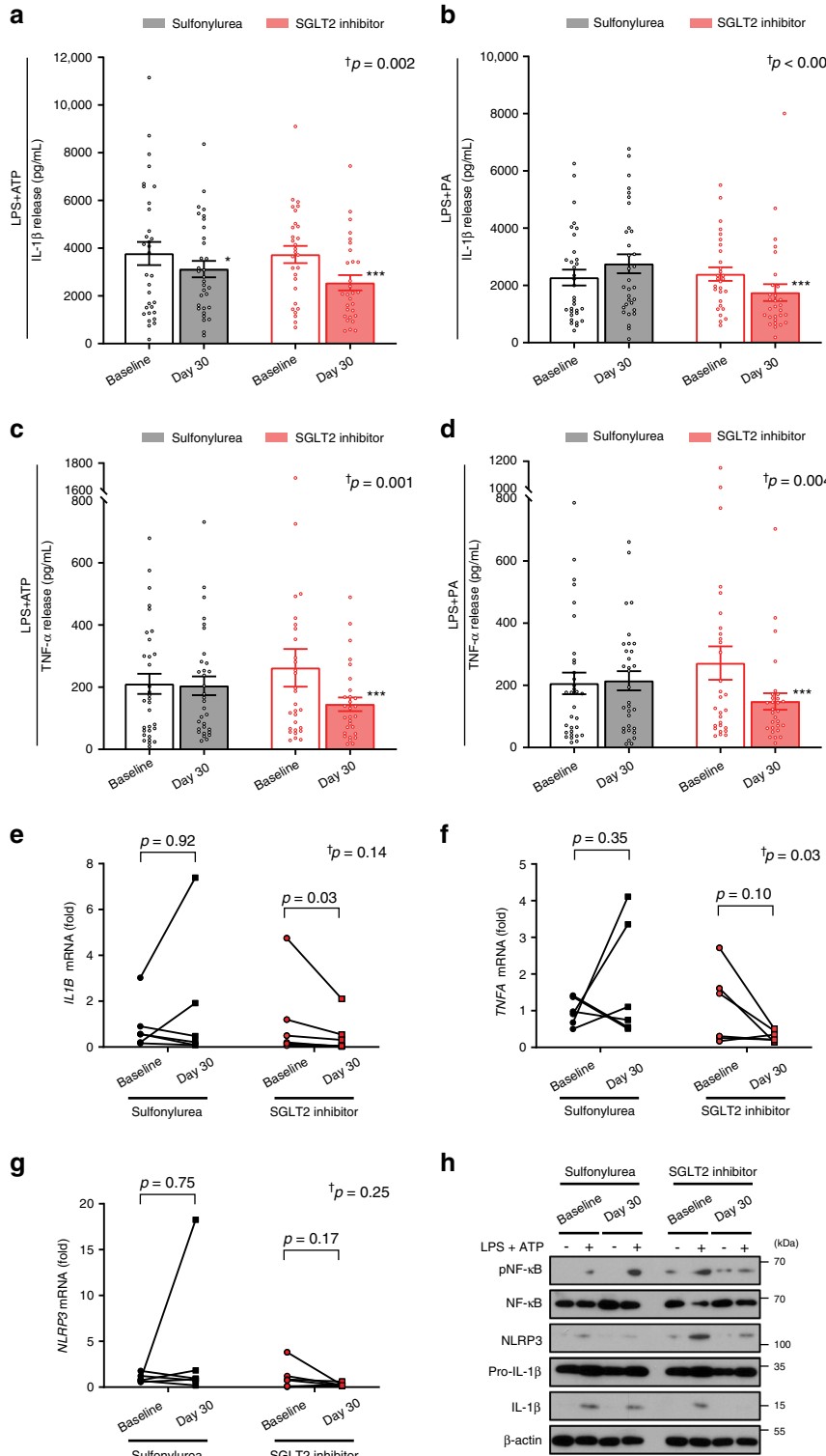

**Fig. 3 Effects of SGLT2 inhibitor and sulfonylurea on secretion of IL-1β and TNF-α from macrophages.** ELISA assay measurement of IL-1β secretion from macrophages exposed to 2 mM ATP (**a**) or 0.2 mM palmitate (**b**) with 0.1 µg/mL LPS priming (sulfonylurea group: n = 32, SGLT2 inhibitor group: n = 29). TNF-α secretion from macrophages exposed to ATP (**c**) or palmitate (**d**). Experiments were repeated twice or three times per sample; bar graphs are drawn using mean values of those results per sample, whereas the statistical significances are derived from raw data. Data are represented as mean ± SEM. Two-sided paired t test; *P < 0.05, **P < 0.01, and ***P < 0.001 versus baseline. mRNA levels encoding *IL1B* (**e**), *TNFA* (**f**), and *NLRP3* (**g**) (n = 6 per group). Two-sided paired t test or Wilcoxon signed rank test. †Statistical significance for the time × group interaction evaluated by repeat-measures analysis of variance (ANOVA) (Non-normally distributed variables were log transformed for analysis and back transformed for presentation). **h** Representative protein levels of molecules regarding NLRP3 inflammasome activation with or without LPS and ATP stimulation. IL-1β interleukin-1β, NF-κB nuclear factor kappa-light-chain-enhancer of activated B cells, NLRP3 NLR family, pyrin domain-containing 3, PA palmitate, SEM standard error of the mean, SGLT2 sodium–glucose cotransporter 2, TNF-α tumor necrosis factor-α. Source data are provided as a Source Data file.

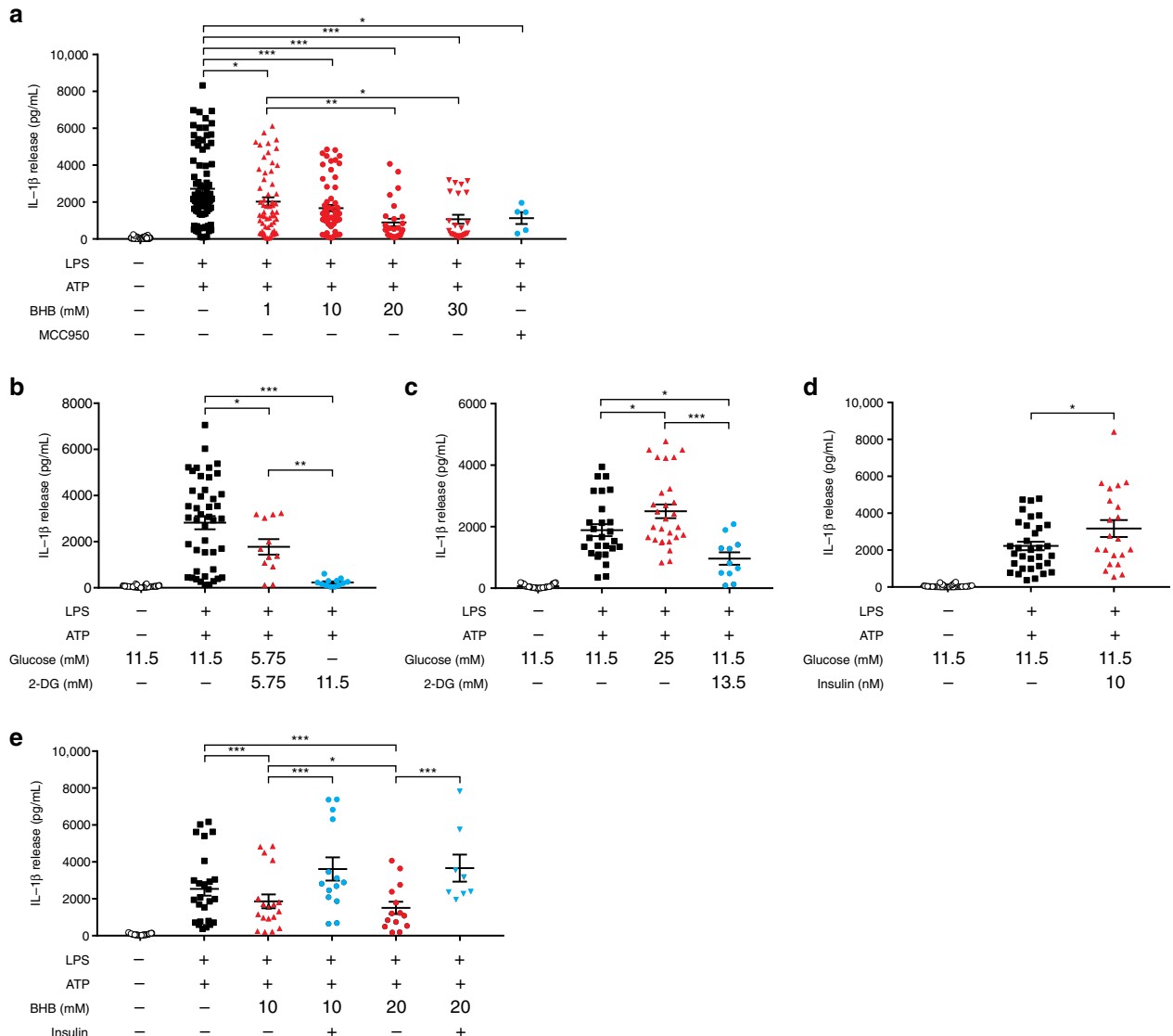

**Fig. 4 Effects of BHB, glucose, and insulin on NLRP3 inflammasome activation in human macrophages by ELISA assay.** IL-1β secretion when exposed to vehicle or 2 mM ATP with 0.1 μg/mL LPS priming and increased BHB (1, 10, 20, and 30 mM) or 100 nM MCC950, a small-molecule inhibitor of NLRP3 inflammasome (administered for the last 5 hours as a positive control) (**a**); increased 2-DG (**b**); increased glucose (11.5 mM vs. 25 mM) (**c**); 11.5 mM glucose with/without 10 nM insulin (**d**); increased BHB (10 mM vs. 20 mM) with/without 10 nM insulin (**e**). 5–64 biologically independent samples per treatment; experiments on each treatment were repeated up to three times per sample. Symbols are data points from independent experiments: n = 137, 90, 58, 64, 28, 24, 5 (**a**, left to right), 66, 44, 12, 12 (**b**, left to right), 39, 27, 28, 11 (**c**, left to right), 48, 33, 21 (**d**, left to right), 39, 26, 18, 14, 14, 8 (**e**, left to right). Data are represented as mean ± SEM. One-way analysis of variance (ANOVA) using Tukey's test or a two-tailed Student's *t* test with the Bonferroni method for adjusting *P* values; *P < 0.05, **P < 0.01, and ***P < 0.001. BHB β-hydroxybutyrate, IL-1β interleukin-1β, NLRP3 NLR family, pyrin domain-containing 3, SEM standard error of the mean, 2-DG 2-deoxyglucose. Source data are provided as a Source Data file.

**Isolation and culture of PBMCs.** Samples of whole blood were collected into acid citrate dextrose tubes. PBMCs were isolated from blood by density centrifugation (20 min at 1600 × g (without brakes) at 18–20 °C) using Ficoll Medium (Ficoll-Paque PLUS, GE Healthcare Life Science, Uppsala, Sweden)[13]. After removing the top layer of clear plasma, the PBMC-containing layer was aspirated and the cells were washed with Dulbecco's phosphate-buffered saline. Then the cells were re-suspended in RPMI-1640 supplemented with 1% penicillin, 1% streptomycin, and 10% fetal bovine serum. To generate human macrophages, cells were incubated at 1 × 10⁶ cells per mL in 24-well plates in RPMI medium plus 10% fetal bovine serum for 2 h and then incubated with 20 ng/mL M-CSF for 3 days. After 3 days, the cells were then incubated with fresh RPMI medium plus 10% fetal bovine serum containing 10 ng/mL M-CSF and the medium was freshly replaced every 2 days (total 7 days) as previously described[13].

**Cell stimulation and cytokine assays.** Human macrophages were incubated at 1 × 10⁶ cells per mL in 24-well plates in RPMI medium plus 10% fetal bovine serum with 0.1 μg/mL LPS (Sigma-Aldrich, L6529) for 4 h[13]. To stimulate the release of IL-1β,

2 mM ATP (Sigma-Aldrich) or 0.2 mM palmitate (Sigma-Aldrich, P9767) was added for the last 1 or 12 h of incubation, respectively. In addition, to evaluate the direct inhibitory or stimulating effects on NLRP3 inflammasome activation, different concentrations of BHB (1, 10, 20, and 30 mM), glucose (11.5 and 25 mM) and 2-DG as a calorie restriction mimetic, or insulin (10 nM) were pre-treated for 5 h before adding LPS and ATP. Supernatants were collected, centrifuged to remove cells and debris, and stored at −80 °C for later analysis. IL-1β and TNF-α was measured using ELISA (eBioscience, human 88-7261-88 and human 88-7346-88, respectively). Results were normalized to cell number, as determined by the CyQuant cell proliferation assay (Invitrogen). Experiments on the participants were repeated up to three times per sample. The examiner conducting these experiments remained blinded to the subject's clinical status and treatment group assignment throughout the study.

**Immunoblot analysis.** Total cell lysates with or without stimulation by LPS and ATP were prepared by lysis of human macrophages with RIPA buffer (cell signaling, 9806) (20 mM Tris-HCl, 150 mM NaCl, 1% NP-40, 1% sodium deoxycholate,

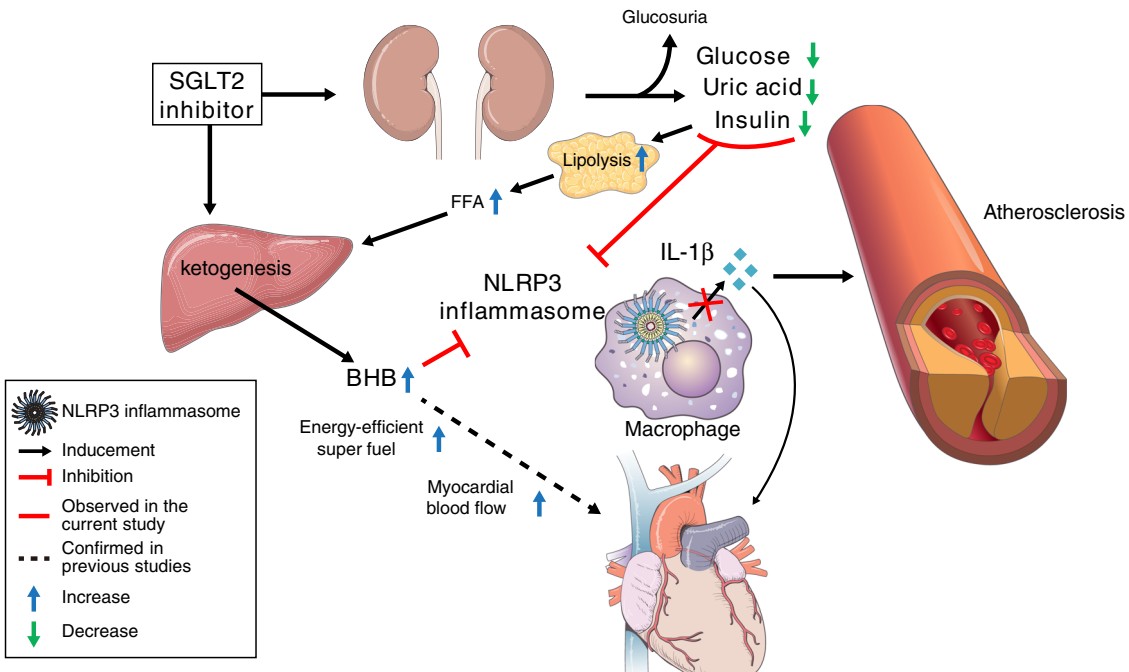

**Fig. 5 Scheme representing the proposed effects of SGLT2 inhibitor on NLRP3 inflammasome activation.** BHB β-hydroxybutyrate, FFA free fatty acid, IL-1β interleukin-1β, NLRP3 NLR family, pyrin domain-containing 3, SGLT2 sodium–glucose cotransporter 2.

2.5 mM sodium pyrophosphate) and the protein contents were measured using the Bradford assay (Bio-Rad, 500-0006). Equivalent amounts of each protein extract were heat denatured in 5× sample buffer (2% sodium dodecyl sulfate, 62.5 mM Tris (pH 6.8), 0.01% bromophenol blue, 1.43 mM mercaptoethanol, and 0.1% glycerol), separated on 10% polyacrylamide gels, and electrophoretically transferred onto a polyvinylidenefluoride membrane (Bio-Rad, 1620175). After blocking, membranes were treated with the following antibodies: anti-p-nuclear factor kappa-light-chain-enhancer of activated B cells (pNF-κB) (Ser536; Cell Signaling Technology, 3033), anti-NF-κB (Cell Signaling Technology, 8242), anti-NLRP3 (Santa Cruz Biotechnology, sc-66846), and anti-IL-1β (Santa Cruz Biotechnology, sc-7884). Immunostaining was performed using chemiluminescent reagents (SuperSignal West Pico Luminol/Enhancer solution; Thermo Scientific, 34080) and Agfa medical X-ray film (Mortsel, CURIX 60). Actin protein levels were used as a loading control. Uncropped scans for western blot analysis are shown in the Source Data file.

**Quantitative PCR analysis.** Total RNA was isolated from cells with TRIzol reagent (Invitrogen, 15596–018) following the manufacturer's instructions, and then 2 μg total RNA was reverse transcribed into complementary DNA (cDNA) using the High Capacity cDNA Reverse Transcription kit (Applied Biosystems, 4368814). The cDNA was then amplified in the ABI 7500 sequence detection system (Applied Biosystems, 4350584) using Power SYBR® Green PCR Master Mix (Applied Biosystems, 4367659) with the following cycling conditions: 40 cycles of 95 °C for 5 s, 58 °C for 10 s, and 72 °C for 20 s. Target gene expression was normalized to that of glyceraldehyde 3-phosphate dehydrogenase (*Gapdh*) or *Actin*, and quantitative analyses were conducted using the ΔΔcycle threshold method and StepOne Software version 2.2.2. The primer sets used for qPCR were as follows: *IL1B*, forward 5′-GGA CAA GCT GAG GAA GAT GC-3′ and reverse 5′-TCG TTA TCC CAT GTG TCG AA-3′; *TNFA*, forward 5′–GTG ACA AGC CTG TAG CCC AT-3′ and reverse 5′–TAT CTC TCA GCT CCA CGC CA-3′; *NLRP3*, forward 5′–TCT CAC GCA CCT TTA CCT GC-3′ and reverse 5′–GAT CCC AGC AGC AGT GTG AC-3′; *β–actin*, forward 5′-GGA CTT CGA GCA AGA GAT GG-3′ and reverse 5′-AGC ACT GTG TTG GCG TAC AG-3′.

**RNA extraction and sequencing.** Total RNA was isolated from macrophage lysates obtained from PBMCs of seven individuals (three from the sulfonylurea arm, four from the SGLT2 inhibitor arm) before and after treatment using a Qiagen RNA extraction kit (Qiagen, Valencia, CA, USA). Among seven individuals, three in sulfonylurea group and 3 of 4 in SGLT2 inhibitor group were treated with metformin before and throughout the study period. Total RNA quality and quantity was verified on a NanoCrop1000 spectrometer (Thermo Scientific, Wilmington, DE, USA) and Bioanalyzer 2100 (Agilent technologies, Palo Alto, CA, USA). RNA sequencing was performed by Macrogen (Seoul, Korea). Libraries were constructed with the TruSeq RNA Access Library Prep kit (Illumina, San Diego, CA, USA) and the enriched library was sequenced on an Illumina HiSeq

4000 system. In brief, we started with 1 μg of total RNA, mRNA was first purified using polyA selection, then chemically fragmented and converted into single-stranded cDNA using random hexamer priming. Next, the second strand was generated to create double-stranded cDNA that was ready for the TruSeq library construction. The short ds-cDNA fragments were then connected with sequencing adapters, and suitable fragments were separated by agarose gel electrophoresis. Finally, TruSeq strand-specific mRNA libraries were built by PCR amplification and quantified using qPCR according to the qPCR Quantification Protocol Guide and qualified using Bioanalyzer 2100 (Agilent technologies, Palo Alto, CA, USA).

**Bioinformatics of RNA-seq data.** Sequence reads were mapped against the human reference genome (NCBI hg19) and analyzed using CLC Genomics Workbench v.9.0.1 software (CLC Bio, Cambridge, MA, USA). Among a total of 57,773 genes annotated for expression values based on transcripts, differentially expressed genes among two groups with statistical significance (*P* value < 0.05) using the Baggerley's test were selected for further analysis. Afterwards, fold change calculated as mean RPKM values of drug group divided by mean RPKM values of control group (pre-treatment samples) was utilized for further filtering step with cut-off value of 1.3 for upregulation and 0.77 for downregulation.

**GO enrichment analysis.** GO enrichment analyses were carried out using database for annotation, visualization, and integrated discovery (DAVID) (https://david.ncifcrf.gov/). ENSEMBL gene IDs were used instead of official gene symbols. The background species was selected as Homo Sapiens for genes with upregulation and downregulation values. Annotation of genes into GO terms with functional categories, general annotations, and protein interactions was executed, and biological process was used for GO category. Afterwards, GO terms with statistical significance (Benjamini corrected *P* value < 0.05) were further visualized using reduce visualize gene ontology (REViGO) (http://revigo.irb.hr/) using a cut-off of 0.7 for SimRel similarity scores. Finally, we determined major representative GO terms and grouped all GO terms into clusters according to treemap view. In addition, individual GO terms with statistical significance were ranked by degree of statistical significance according to values of –log(Benjamini corrected *P* value).

**Statistical analysis.** All statistical analyses were performed using SPSS version 23.0 for Windows (IBM Corp., Armonk, NY, USA). A normality test was performed for all continuous variables. The characteristics of the study participants were analyzed according to groups using a Mann–Whitney U or two-sample Student's *t* test for continuous variables and a Pearson $\chi^2$ test for categorical variables. The effects of empagliflozin or glimepiride treatment on secretion of IL-1β and TNF-α from macrophages and metabolic parameters were assessed by two-sided paired *t*-test or Wilcoxon signed rank test. Statistical significance for the time × group interaction was evaluated by using repeat-measures analysis of

variance (ANOVA). Non-normally distributed variables were log transformed for analysis and back transformed for presentation. The Pearson's correlation coefficient was used for correlation. One-way ANOVA using Tukey's test or a two-tailed Student's $t$ test with the Bonferroni method for adjusting $P$ values for the number of comparisons being made were used to examine differences between treatments in ex vivo experiments. All $P$ values $< 0.05$ were considered statistically significant.

**Reporting summary**. Further information on research design is available in the Nature Research Reporting Summary linked to this article.

## Data availability

All data generated or analyzed during this study are included in this published article and its Supplementary Information file. The source data underlying Tables 1, 2, Figs. 2–4, and Supplementary Table 2 are provided as a Source Data file. The RNA-seq data are available at the NCBI GEO database with the accession number GSE134329 [https://www.ncbi.nlm.nih.gov/geo/query/acc.cgi?acc=GSE134329].

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

## Acknowledgements

The authors would like to thank Dong-Su Jang, MFA, (Medical Illustrator) for his help with producing the illustration (Fig. 5). Editorial assistance was provided by Caron Modeas. This work was supported by the National Research Foundation of Korea (NRF) Grant [NRF-2016R1A5A1010764 and NRF-2017R1C1B5015044] funded by the Korean Government (MSIP), the grant of the Korea Health technology R&D Project through the Korea Health Industry Development Institute (KHIDI) funded by the Ministry of Health & Welfare, Republic of Korea [HI17C0913], and the grant of the Industry Technology Development Program [10063335] funded by the Ministry of Trade, Industry and Energy (MOTIE, Korea). It was also supported by a faculty research grant of Yonsei University College of Medicine for [6-2016-0082 to J.-S.K].

## Author contributions

S.R.K. and S.-G.L. wrote the manuscript, analyzed the data, and performed the statistical analysis, S.H.K., J.H.K., E.C., and I.H. conducted the experiments, W.C., M.L., and C.-M.O. contributed to acquisition of data, J.H.R. and H.Y.G. performed the bioinformatic analyses, C.J.L. analyzed the data and performed the statistical analysis, J.Y.J., J.-H.K., B.-W.L., E.S.K, B.-S.C., M.-S.L., J.-W.Y., and J.W.C. provided critical review, advice, and consultation throughout. J.-S.K. and Y.-h.L. contributed to the conception and design of the study, the interpretation of data, and the drafting and critical revision of the manuscript.

## Competing interests

The authors declare no competing interests.
