## [Peer Review File · Nature Communications]

Editorial Note: Parts of this Peer Review File have been redacted as indicated to maintain the confidentiality of unpublished data removed from the manuscript during peer review due to editorial policies.

Reviewers' comments:

Reviewer #1 (Remarks to the Author):

So Ra Kim et al

SGLT2 inhibition modulates NLRP3 inflammasome activity in patients with type 2 diabetes and cardiovascular disease

The authors assessed the effect of SGLT2 inhibitor empagliflozin on NLRP3 inflammasome activity in patients with type 2 diabetes and high cardiovascular risk who received either an SGLT2 (n=29) inhibitor or a sulfonylurea (n=32) for 30 days. The paper describes the many metabolic changes observed and particularly draws attention to the changes of the NLRP3 inflammasome activation analyzed in macrophages. [REDACTED].

1. The paper is well-written, the statistics are robust and appropriate and the descriptions of the methods generally good. I have no major criticisms.
2. I applaud the number and detail of the investigations in what is both a randomized trial and a physiological investigation.
3. The risk, however, which is not outlined in the limitations, is that with very many metabolic measurements it is possible to conclude a multiplicity of 'pathways' which may be, among other possibilities, direct, indirect, or even agent-specific. So the hypothesis has to be tentative and in deference to the careful writing of the authors I acknowledge that they use the terms 'proof of principle' and their bottom line 'The present data suggest that these mechanisms might help to explain the cardioprotective effects of SGLT2 inhibitor in humans' is appropriate language.
4. [REDACTED]
5. The ethical basis and the trial registration for the main comparison is mentioned – "This study complied with the Declaration of Helsinki, and is registered at clinicaltrials.gov. (NCT02964572). The protocol was approved by the Institutional Review Board at Severance Hospital (04-2016-0795). All participants provided their written informed consent." [REDACTED].

Prof David R. Matthews.

Reviewer #2 (Remarks to the Author):

This study by Lee and colleagues was initiated to investigate why SGLT2 inhibitors reduce adverse CVD events in T2D patients and whether this effect was independent of its glucose-lowering effects. It's an interesting study that reveals an important finding that SGLT2 inhibitors may have additional clinical benefits lowering NLRP3 inflammasome activation, which itself has been linked to many chronic inflammatory diseases. Given SGLT inhibitors increase BHB the authors tested in the involvement of this metabolite in possible regulation of the inflammation via the inflammasome pathway. [REDACTED]. Overall this is an well done study that provides new insights on the clinical relevance of SGLT2 mediated ketogenesis on inflammasome activation.

Following points should be addressed –

1. Biochemical caspase-1 cleavage in myeloid cells has not been shown and if possible should be included. If authors do not have samples from the trial, is it possible to study the mechanism of SGLT2 inhibitor Vs BHB in vitro from cells of healthy Vs diabetic donors?
2. The prior papers have shown that BHB also inhibits the IL1B secretion in human

monocytes and neutrophils. Please discuss the mechanism of effects of SGLT2 inhibitors [REDACTED] on serum IL-18 and IL1B?

3. It is stated that some subjects were already taking Metformin and were allowed to continue this medication throughout the trial. However, it is not particularly clear who these subjects were and whether these subjects were used in any of the assays in which only a subset of participants were used (importantly RNAseq). This is critical given that Metformin likely has NLRP3-altering capabilities due to its AMPK-stimulating actions.

4. SGLT2 group lost more weight. Is weight-loss the biggest driver of observations of reduced inflammation? Please discuss.

Reviewer #3 (Remarks to the Author):

In their paper «SGLT2 inhibition modulates NLRP3 inflammasome activity in patients with type 2 diabetes and cardiovascular disease» Kim et al. evaluate in vivo in a randomized manner in a relatively small cohort of patients well matched patients the effects of empagliflozin and glimepiride for 30 days on inflammatory pathways.

This design and the hypothesis that interleukin-1beta might be involved is novel, interesting and clinically potentially important. The measured multiple parameters in plasma of the patients and in monocytes obtained from the recruited subjects.

They found that both drugs similarly lowered plasma glucose, but empagliflozin led to a greater reduction of interleukin-1 beta.

Empagliflozin, but not glimepiride led to a marked increase in plasma levels of fasting beta-hydroxy butyrate and a significant decrease in uric acid and fasting insulin, while free fatty acids increased. Furthermore, empagliflozin improved insulin sensitivity, while glimepirides stimulated insulin secretion as expected.

This study is novel, interesting and well designed and written and illustrated with appropriate figures summarizing the data.

There are a few downsides, however:

1. The mechanism by which empagliflozin would activate the inflammasome NLRP3 is not addressed. NLRP3 is activated by crystals, among them cholesterol and uric acid. As uric acid changes, could this be involved? Additional data should be provided on this aspect.
2. In the ex vivo experiments NLRP3 only tended to be reduced by empagliflozin, while interleukin-1beta was significantly reduced – is there thus another mechanism involved rather than through NLRP3? Although RNA sequencing provided insights, this is not properly followed up.
3. Would empagliflozin in vitro still reduce interleukin-1beta secretion or expression in macrophages, if NLRP3 expression would be silenced beforehand?
4. Although beta-hydroxybutyrate has been described to suppress NLRP3 in mice, what is the mechanism in human macrophages?

Response to Reviewers' Comments

We would like to thank the reviewers for their comments and criticisms, which helped to greatly improve the quality of this manuscript.

Reviewers' comments:

Reviewer #1 (Remarks to the Author):

So Ra Kim et al

SGLT2 inhibition modulates NLRP3 inflammasome activity in patients with type 2 diabetes and cardiovascular disease

The authors assessed the effect of SGLT2 inhibitor empagliflozin on NLRP3 inflammasome activity in patients with type 2 diabetes and high cardiovascular risk who received either an SGLT2 (n=29) inhibitor or a sulfonylurea (n=32) for 30 days. The paper describes the many metabolic changes observed and particularly draws attention to the changes of the NLRP3 inflammasome activation analyzed in macrophages.

[REDACTED].

Comment 1: The paper is well-written, the statistics are robust and appropriate and the descriptions of the methods generally good. I have no major criticisms.

Comment 2: I applaud the number and detail of the investigations in what is both a randomized trial and a physiological investigation.

Comment 3: The risk, however, which is not outlined in the limitations, is that with very many metabolic measurements it is possible to conclude a multiplicity of 'pathways' which may be, among other possibilities, direct, indirect, or even agent-specific. So the hypothesis has to be tentative and in deference to the careful writing of the authors I acknowledge that they use the terms 'proof of principle' and their bottom line 'The present data suggest that these mechanisms might help to explain the cardioprotective effects of SGLT2 inhibitor in

humans' is appropriate language.

Response: We deeply respect your comprehensive understanding of our research. As the reviewer commented, based on well-established evidences of the critical contribution of the NLRP3 inflammasome/IL-1 β to cardiovascular (CV) diseases, our study is a 'proof-of-concept' randomized controlled trial aimed to investigate the potential mechanism of SGLT2 inhibitors regarding CV protection and regulation of inflammasome activity.

Based on your comments, the limitations have been revised as follows:

(Discussion)

"A limitation of this study is that we did not assess whether the changes in inflammasome activity by SGLT2 inhibitor could be linked to the improvement in CV outcomes The present study is a proof-of-concept RCT aimed to elucidate the glucose-independent mechanism of SGLT2 inhibitors regarding CV protection. *Besides our findings, it is important to bear in mind that there may be multiplicity of other pathways or mechanisms which can be directly or indirectly involved in the protection of CVD by SGLT2 inhibitors.* Nevertheless, this is the first study to provide evidence that SGLT2 inhibitor suppresses NLRP3 inflammasome activation in patients with T2D at high risk of CVD."

Comment 4: [REDACTED].

Response: Thank you for the valuable comments. The Title and the Abstract have been revised as follows:

(Title)

"SGLT2 inhibition modulates NLRP3 inflammasome activity via changes in ketones and insulin in type 2 diabetes and cardiovascular disease"

(Abstract)

[REDACTED]

Comment 5: The ethical basis and the trial registration for the main comparison is mentioned – “This study complied with the Declaration of Helsinki, and is registered at clinicaltrials.gov.(NCT02964572). The protocol was approved by the Institutional Review Board at Severance Hospital (04-2016-0795). All participants provided their written informed consent.”[REDACTED]

Response: Thank you for the important comments. [REDACTED]

The manuscript has been revised to include:

(Methods)

[REDACTED]

Reviewer #2 (Remarks to the Author):

This study by Lee and colleagues was initiated to investigate why SGLT2 inhibitors reduce adverse CVD events in T2D patients and whether this effect was independent of its glucose-lowering effects. It's an interesting study that reveals an important finding that SGLT2 inhibitors may have additional clinical benefits lowering NLRP3 inflammasome activation, which itself has been linked to many chronic inflammatory diseases. Given SGLT inhibitors increase BHB the authors tested in the involvement of this metabolite in possible regulation of the inflammation via the inflammasome pathway. [REDACTED]. Overall this is an well done study that provides new insights on the clinical relevance of SGLt2 mediated ketogenesis on inflammasome activation.

Following points should be addressed –

Comment 1: Biochemical caspase-1 cleavage in myeloid cells has not been shown and if possible should be included. If authors do not have samples from the trial, is it possible to study the mechanism of SGLT2 inhibitor Vs BHB in vitro from cells of healthy Vs diabetic donors?

Response: Thank you for the valuable comments. Unfortunately, we did not have residual samples of myeloid cells from the trial to explore the cleavage of caspase-1. Thus, we obtained PBMCs from diabetic donors and examined caspase-1 activation using western blots that detect the enzymatically active p20 subunit of caspase-1. It was really difficult to demonstrate biochemical cleavage of caspase-1 using human samples. As a result, BHB (20 mM) inhibited both the ATP- induced cleavage of caspase-1 into p20 and the processing of the biologically active form of IL-1 β , whereas treatment in vitro with SGLT2 inhibitor empagliflozin (10 μ M) did not inhibit caspase-1 activation (as shown in the figure below). Given that the maximum plasma concentration of empagliflozin is less than 1 μ M when treated with 25 mg empagliflozin once daily in patients with type 2 diabetes¹, it seems that empagliflozin does not directly inhibit NLRP3 activation.

Fig. Rev. 1. Western blot analysis of caspase-1 and IL-1β in human macrophages primed with LPS and stimulated with ATP in the presence of BHB or empagliflozin.

Isolation of PBMCs from whole blood and differentiation into macrophages were performed as described in the main manuscript. Immunoblots from PBMCs untreated (Unt) or treated with BHB (10 or 20 mM), empagliflozin (10 μM) and LPS (0.5 μg/mL) for 3 h, followed by treatment with ATP (3 mM) for 1 h, as indicated. Cells were lysed in buffer containing 20 mM HEPES (pH7.5), 0.5 % Nonidet P-40, 50 mM KCl, 150 mM NaCl, 1.5 mM MgCl₂, 1 mM EGTA, and protease inhibitors. Samples were then centrifuged at 20,000 g for 12 min, the supernatant was collected and the protein contents were measured using the Bradford assay (Bio-Rad, 161-0158). Equivalent amounts of each protein extract were heat-denatured in 5 x sample buffer (0.3 M Tris-Cl (pH6.8), 500 mM DTT, 10 % SDS, 50 % Glycerol, 0.05 % Bromophenol blue). Cell culture supernatants were precipitated by the addition of an equal volume of methanol and 0.25 volumes of chloroform, then were vortexed and centrifuged for 10 min at 20,000 g. The upper phase was discarded and 500 μl methanol was added to the interphase. This mixture was centrifuged for 10 min at 20,000 g and the protein pellet was dried at 55 °C, resuspended in 2 x sample buffer and boiled for 8 min at 98 °C. Samples separated on 10% polyacrylamide gels, and electrophoretically transferred onto a polyvinylidene fluoride membrane (Bio-Rad, 162-0177). After blocking, membranes were treated with the following antibodies: anti-IL-1β (Cell

Signaling Technology, 12703S) and anti-caspase-1 (Cell Signaling Technology, 3866S). Immunostaining was performed using chemiluminescent reagents (Clarity Max Western ECL Substrate; Biorad, 170-5062) and Agfa medical X-ray film (CP-G plus). Actin protein levels were used as a loading control.

BHB, β -hydroxybutyrate; EMPA, empagliflozin; Sup, supernatants; Lys, lysates; Procaspl, procaspase-1; IL-1 β , interleukin-1 β .

Taken together, SGLT2 inhibitor might inhibit caspase-1 and NLRP3 activation via several metabolic changes including BHB, rather than having an inhibitory effect in itself.

Comment 2: The prior papers have shown that BHB also inhibits the IL1B secretion in human monocytes and neutrophils. Please discuss the mechanism of effects of SGLT2 inhibitors and [REDACTED] on serum IL-18 and IL1B?

Response: We have measured serum IL-1 β and IL-18 levels with the residual samples of the participants. Serum IL-1 β and IL-18 were measured with quantitative enzyme-linked immunosorbent assay (ELISA) using human Quantikine HS ELISA kits (R&D Systems, MN, USA). The assay sensitivities (the minimum detectable levels) for IL-1 β and IL-18 were 0.033 pg/mL and 1.25 pg/mL, respectively, determined by zero standard +2 standard deviation.

Supplementary Table 2. Effects of sulfonylurea and SGLT2 inhibitor on serum IL-1 β and IL-18 levels.

	Sulfonylurea (n = 32)			SGLT2 inhibitor (n = 29)		
	Before	After	P	Before	After	P
Serum IL-1 β (pg/mL)*	0.17 (0.08-0.53)	0.13 (0.05-0.84)	0.83	0.11 (0.07-0.35)	0.08 (0.06-0.23)	0.29
Serum IL-18 (pg/mL)	227.7 (168.8-283.0)	243.4 (181.2-266.4)	0.54	218.2 (174.4-262.8)	216.6 (164.7-286.9)	0.57

Statistical significance was evaluated by Wilcoxon signed rank test; values are described as median (interquartile range).

*Not detectable (<0.033 pg/mL) in 37 (18 in sulfonylurea group and 19 in SGLT2 inhibitor group) out of 61 participants; excluded from the analyses.

IL-1 β , interleukin-1 β ; IL-18, interleukin-18; SGLT2, sodium-glucose cotransporter 2.

[REDACTED]

As a result, despite a tendency to decrease after SGLT2 inhibitor treatment, there were no significant changes in serum cytokine concentrations [REDACTED]. This may be attributable to the following factors:

First, serum IL-1 β levels are often undetectable even in diseases with clear evidence of increased IL-1 activity²⁻⁵; In our study, 37 out of 61 participants, had serum IL-1 β levels below the detection limits of the assay used (<0.033 pg/mL). In a previous cohort study, although IL-1 β levels were not different among body mass index (BMI) classes, the plasma level of IL-1 receptor antagonist was significantly higher in individuals with high BMI which might be attributable to the longer half-life and more constant levels over time of circulating cytokine receptors than cytokines themselves⁶. The challenges of direct measurement of serum IL-1 β levels might lead to a failure to detect differences in serum IL-1 β levels before and after drug treatment.

Second, extensive use of aspirin or statin (more than 90 %) in both sulfonylurea and SGLT2 inhibitor groups might attenuate changes in serum proinflammatory cytokines. It is well known that aspirin or statin therapy can reduce proinflammatory cytokines and CRP⁷⁻⁹. [REDACTED].

Finally, IL-1 β signaling in the immune cells of vessel tissues or of atherosclerotic lesions

enriched for various endogenous danger signals, rather than circulating IL-1 β levels, might be a major determinant of atheroma formation or heart failure^{5,10,11}. IL-1 β exerts its effect in an autocrine/paracrine fashion and, consequently, the detected levels of IL-1 β in the plasma may be very low¹². Furthermore, although most circulating cytokines are secreted from activated macrophages and lymphocytes, adipocytes and skeletal muscle are also a possible source of cytokines^{13,14}. In addition, activation of the NLRP3 inflammasome requires two independent steps: priming and triggering¹⁵. Therefore, cytokine production capacity may not be related to serum concentration. IL-1 β serum concentration probably represents only an indirect marker of chronic inflammation and a fraction of the real production capacity of activated monocytes/macrophages¹⁶. In the present study, we identified that treatment with an SGLT2 inhibitor can reduce the activation of NLRP3 inflammasome in human macrophages after stimulation with LPS and ATP/palmitate.

The manuscript has been revised to include:

(Results)

“SGLT2 inhibitor suppresses NLRP3 inflammasome activation

....

Regarding serum IL-1 β and IL-18 levels, despite a tendency to decrease after SGLT2 inhibitor treatment, there were no significant changes in those concentrations (Supplementary Table 2). This may be attributable to the following factors: First, serum IL-1 β levels are often undetectable even in diseases with clear evidence of increased IL-1 activity^{5,17-19}. IL-1 β exerts its effect in an autocrine/paracrine fashion and, consequently, the detected levels of IL-1 β in the serum may be very low²⁰; In our study, 37 out of 61 participants, had serum IL-1 β levels below the detection limits of the assay used. Second, extensive use of aspirin or statin in both sulfonylurea and SGLT2 inhibitor groups might attenuate changes in serum proinflammatory cytokines²¹. Finally, as activation of the NLRP3 inflammasome requires two independent steps: priming and triggering²², IL-1 β signaling in the immune cells of vessel tissues or of atherosclerotic lesions enriched for various endogenous danger signals, rather than circulating IL-1 β levels, might be a major determinant of atheroma formation or heart failure^{5,23,24}. In the present study, we identified that treatment with an SGLT2 inhibitor can reduce the activation of

NLRP3 inflammasome in human macrophages after stimulation with LPS and ATP/palmitate.”

(Methods)

“Clinical and laboratory measurements

.... Serum IL-1 β and IL-18 were measured with ELISA using human Quantikine HS ELISA kits (R&D Systems, MN, USA). The assay sensitivities (the minimum detectable levels) for IL-1 β and IL-18 were 0.033 pg/mL and 1.25 pg/mL, respectively, determined by zero standard +2 standard deviation.”

Comment 3: It is stated that some subjects were already taking Metformin and were allowed to continue this medication throughout the trial. However, it is not particularly clear who these subjects were and whether these subjects were used in any of the assays in which only a subset of participants were used (importantly RNAseq). This is critical given that Metformin likely has NLRP3-altering capabilities due to its AMPK-stimulating actions.

Response: The proportion of participants who were administered metformin during the study period was as follows (shown in Table 1 in the main manuscript), and there was no difference between groups.

	Sulfonylurea	SGLT2 inhibitor	P
Metformin user	31/32 (96.9%)	26/29 (89.7%)	0.34

As the use of metformin is first-line therapy, most participants had already been administered metformin and were included in all analyses of the present study. Since in each participant, preexisting metformin was maintained before and throughout the study period in dosage regimen, additional subgroup analyses according to metformin use were not performed. Among 7 subjects of RNA seq analysis, 3 of 3 in sulfonylurea group and 3 of 4 in SGLT2 inhibitor group were treated with metformin before and throughout the study period.

The manuscript has been revised to include:

(Results)

“Baseline characteristics of study participants

.... As metformin is first-line therapy, most participants in both groups were taking it and continued on the same dosage during the study period. ...”

(Methods)

“RNA extraction and sequencing

Total RNA was isolated from macrophage lysates obtained from PBMCs of 7 individuals (3 from the sulfonylurea arm, 4 from the SGLT2 inhibitor arm) before and after treatment using a Qiagen RNA extraction kit (Qiagen, Valencia, CA, USA). *Among 7 individuals, 3 of 3 in sulfonylurea group and 3 of 4 in SGLT2 inhibitor group were treated with metformin before and throughout the study period.* Total RNA quality and quantity was verified on ...”

Comment 4: SGLT2 group lost more weight. Is weight-loss the biggest driver of observations of reduced inflammation? Please discuss.

Response: Thank you for this valuable comment. We conducted repeat-measures ANOVA adjusted for body weight change. As a result, treatment with SGLT2 inhibitor significantly inhibited NLRP3 inflammasome activation, independent of body weight change, compared to sulfonylurea (**time x group interaction $P = 0.02$**). In addition, there was no significant correlation between changes in body weight and changes in IL-1 β or TNF- α release (as shown in the figures below).

The manuscript has been revised to include:

(Results)

“In response to ATP stimulation, IL-1 β secretion was significantly reduced after both SGLT2 inhibitor and sulfonylurea treatment ($3,733 \pm 360$ to $2,549 \pm 320$ pg/mL, $P < 0.001$; and $3,777 \pm 485$ to $3,121 \pm 345$ pg/mL, $P = 0.01$, respectively) (Fig. 3a). However, SGLT2 inhibitor showed a greater reduction in IL-1 β secretion compared

to sulfonylurea (time × group interaction $P = 0.002$), *which remained significant after adjustment for body weight change (time × group interaction $P = 0.02$).*”

“.... To rule out the effect of body weight loss on changes in inflammasome activity, correlation analyses were conducted. As a result, there was no significant correlation between changes in body weight and changes in IL-1 β or TNF- α release (Supplementary Fig. 2).”

Supplementary Fig. 2. Correlation between changes in body weight and changes in IL-1 β (a) and TNF- α (b) release ($n = 61$).

Pearson's correlation coefficient. 0.1 μ g/mL LPS; 2 mM ATP.

Δ Body weight (kg) = [Body weight at end of treatment (kg) – Body weight at baseline (kg)];
 Δ IL-1 β release (pg/mL) = [IL-1 β release at end of treatment (pg/mL) – IL-1 β release at baseline (pg/mL)]; Δ TNF- α release (pg/mL) = [TNF- α release at end of treatment (pg/mL) – TNF- α release at baseline (pg/mL)].

IL-1 β , interleukin-1 β ; TNF- α , tumor necrosis factor- α .

Reviewer #3 (Remarks to the Author):

In their paper «SGLT2 inhibition modulates NLRP3 inflammasome activity in patients with type 2 diabetes and cardiovascular disease» Kim et al. evaluate in vivo in a randomized manner in a relatively small cohort of patients well matched patients the effects of empagliflozin and glimepiride for 30 days on inflammatory pathways.

This design and the hypothesis that interleukin-1 β might be involved is novel, interesting and clinically potentially important. The measured multiple parameters in plasma of the patients and in monocytes obtained from the recruited subjects.

They found that both drugs similarly lowered plasma glucose, but empagliflozin led to a greater reduction of interleukin-1 β .

Empagliflozin, but not glimepiride led to a marked increase in plasma levels of fasting β -hydroxybutyrate and a significant decrease in uric acid and fasting insulin, while free fatty acids increased. Furthermore, empagliflozin improved insulin sensitivity, while glimepiride stimulated insulin secretion as expected.

This study is novel, interesting and down sides, however well designed and written and illustrated with appropriate figures summarizing the data.

There are a few down sides, however:

Comment 1: The mechanism by which empagliflozin would activate the inflammasome NLRP3 is not addressed. NLRP3 is activated by crystals, among them cholesterol and uric acid. As uric acid changes, could this be involved? Additional data should be provided on this aspect.

Response: Thank you for the important comments. As SGLT2 is not expressed in macrophages, metabolic conditions induced by empagliflozin might modulate NLRP3 inflammasome activity. Previously, it is reported that β -hydroxybutyrate (BHB) can block the activity of NLRP3 inflammasome¹⁷ and insulin can induce the activity of NLRP3 inflammasome¹⁸ (as shown in the figures below).

[redacted]

Consisted with these findings, we demonstrated that both ex vivo experiments [REDACTED] that **higher BHB levels and lower levels of insulin and glucose** (metabolic conditions after the treatment of empagliflozin) significantly suppressed the activation of the NLRP3 inflammasome (Fig. 4 and 5 in the main manuscript).

As the reviewer commented, it is also established that uric acid crystals activate the NLRP3 inflammasome. SGLT2 inhibitors have been shown to reduce serum uric acid levels through increasing renal clearance of uric acid¹⁹.

We performed additional correlation analyses of the changes in serum uric acid, insulin, and BHB levels and changes in IL-1 β release in all participants (N=61).

Fig. Rev. 2. Correlation between changes in serum uric acid (a), insulin (b), or BHB (c) levels and changes in IL-1 β release ($n = 61$).

Pearson's or Spearman's correlation coefficient. Some variables were log transformed for analysis. 0.1 $\mu\text{g/mL}$ LPS; 0.2 mM PA.

$\Delta \text{Serum uric acid (mg/dL)} = [\text{Serum uric acid at end of treatment (mg/dL)} - \text{Serum uric acid at baseline (mg/dL)}]$; $\Delta \text{Ln (Fasting serum insulin)} = [\text{Ln (Fasting serum insulin at end of treatment (}\mu\text{U/mL}) - \text{Ln (Fasting serum insulin at baseline (}\mu\text{U/mL})]$; $\Delta \text{Ln (Fasting serum BHB)} = [\text{Ln (Fasting serum BHB at end of treatment (mM)} - \text{Ln (Fasting serum BHB at baseline (mM})]$; $\Delta \text{Ln (IL-1}\beta \text{ release)} = [\text{Ln (IL-1}\beta \text{ release at end of treatment (pg/mL)} - \text{Ln (IL-1}\beta \text{ release at baseline (pg/mL})]$. BHB, β -hydroxybutyrate; IL-1 β , interleukin-1 β ; PA, palmitate.

As a result, larger reduction in insulin (as shown in the figure above; b) or increase in BHB (c) levels were correlated with higher reductions in IL-1 β secretion, whereas the changes in serum uric acid levels were not significantly associated with changes in IL-1 β release (a).

Therefore, although SGLT2 inhibitor caused a significant decrease in serum uric acid in the present study (Fig. 2c in the main manuscript), whether this reduction directly contributed to the suppression of NLRP3 inflammasome is not conclusive.

Additional text has been added as follows:

(Results)

“To further evaluate metabolic factors associated with changes in inflammasome activity, correlation analyses were conducted. Changes in fasting serum insulin or BHB levels were significantly correlated with changes in NLRP3 inflammasome activity (Supplementary Fig. 1).”

Supplementary Fig. 1. Correlation between changes in fasting serum insulin (a) or BHB (b) levels and changes in IL-1β release (n = 61).

Pearson's correlation coefficient. Variables were log transformed for analysis. 0.1 μg/mL LPS; 0.2 mM PA.

$\Delta \text{Ln (Fasting serum insulin)} = [\text{Ln (Fasting serum insulin at end of treatment (}\mu\text{U/mL})} - \text{Ln (Fasting serum insulin at baseline (}\mu\text{U/mL})}]$; $\Delta \text{Ln (Fasting serum BHB)} = [\text{Ln (Fasting serum BHB at end of treatment (mM)} - \text{Ln (Fasting serum BHB at baseline (mM})]$; $\Delta \text{Ln (IL-1}\beta \text{ release)} = [\text{Ln (IL-1}\beta \text{ release at end of treatment (pg/mL)} - \text{Ln (IL-1}\beta \text{ release at baseline (pg/mL})]$.

BHB, β-hydroxybutyrate; IL-1β, interleukin-1β; PA, palmitate.

Comment 2: In the ex vivo experiments NLRP3 only tended to be reduced by empagliflozin, while interleukin-1β was significantly reduced – is there thus another mechanisms

involved rather than through NLRP3? Although RNA sequencing provided insights, this is not properly followed up.

Response: Thank you for the valuable comments. As you mentioned, the mRNA and protein levels of NLRP3 were not significantly decreased by SGLT2 inhibitor treatment (Fig. 3g, h in the main manuscript), whereas IL-1 β release was significantly reduced. This indicates that SGLT2 inhibitor treatment might block the secretion of IL-1 β via downstream of NLRP3 signaling. Previous papers demonstrated that IL-1 β can be blocked through various mechanisms even though NLRP3 expression was maintained²⁰. Torin1, a competitive mTORC1/2 inhibitor, completely suppressed caspase-1 and IL-1 β activation in mice bone-marrow-derived macrophages in response to LPS and ATP while NLRP3 expression was unchanged (as shown in the figures below).

[redacted]

- In the present study, we concluded that SGLT2 inhibitor treatment suppresses not NLRP3 inflammasome itself, but the activation of the NLRP3 inflammasome, via combined metabolic effects of increased serum BHB levels and decreased serum levels of glucose and insulin. In a previous paper¹⁷, **BHB** blocks NLRP3 inflammasome activation both by controlling an unknown upstream event that reduces K⁺ efflux from macrophages and by inhibiting ASC polymerization, speck formation and assembly of the inflammasome (as shown in the figure below).

[redacted]

Furthermore, in mouse models expressing knocked-in human NLRP3 with gain-of-function mutations that mimic systemic inflammatory diseases (like Muckle-Wells syndrome (MWS) or familial cold autoinflammatory syndrome (FCAS)), treatment with BHB dose-dependently inhibited IL-1 β secretion (see the figures below) regardless of NLRP3 inflammasome expression, suggesting that BHB also exhibit an inhibitory effect on constitutive NLRP3 inflammasome activation.

[redacted]

As the reviewer mentioned regarding the mechanism, additional text has been added in Results, as follows:

(Results)

“Effects of BHB, glucose, and insulin on NLRP3 inflammasome activation in human macrophages

Although it has been reported that BHB blocks activation of the NLRP3 inflammasome-IL-1 β process *by preventing K⁺ efflux and reducing apoptosis-associated speck-like protein containing a caspase recruitment domain (ASC) oligomerization and speck formation¹², ...”*

- Another putative mechanism is changes in glucose and insulin levels after treatment of SGLT2 inhibitor. Regarding **glucose** and **insulin**'s effects on NLRP3 inflammasome activation, insulin reinforces a pro-inflammatory state and stimulates macrophages to produce IL-1 β (as shown in the figure below; a) via phosphorylation of the kinase AKT (e), upregulation of the gene encoding the GLUT1 (g) and hexokinase 2 (h), increased glycolytic activity (i), and the subsequent production of reactive oxygen species¹⁸ (*This issue has been mentioned in the Discussion section*).

[redacted]

Therefore, SGLT2 inhibitor treatment might suppress signaling pathway of NLRP3 inflammasome activation, rather than reduce NLRP3 itself.

- We agree with the reviewer's concern about RNA seq data. Our GO data confirmed that patients treated with SGLT2 inhibitor had lower expression of immune process-related genes compared to those treated with sulfonylurea. However, because of the small number of samples available for RNA seq and the heterogeneity of study population due to the nature of human trial, there is a limitation to conduct in-depth analysis of upstream transcription factors or epigenetic regulators that may govern the expression of those genes after SGLT2 inhibitor treatment. Based on the reviewer's recommendation, we have conducted additional analysis of RNA sequencing data to find candidate genes associated with IL-1 β secretion. As a result, *SLC7A5* and *IRF1* genes tended to be downregulated after treatment compared to baseline in SGLT2 inhibitor group (as shown in the figures below):

Fig. Rev. 3. RNA sequencing for genes associated with IL-1 β secretion between sulfonylurea vs. SGLT2 inhibitor treatment group.

In previous studies, the role of SLC7A5 and IRF1 in IL-1 β secretion from immune cells was revealed as follows:

- Activation-induced SLC7A5 (a major transporter for essential amino acids) expression in human monocytes and macrophages mediates leucine influx leading to enhanced mTORC1-mediated glycolysis, thereby causing augmented production of IL-1 β (as shown in the figure below; left). Silencing of SLC7A5 led to a significant reduction of IL-1 β secretion (Right).²¹

[redacted]

- Transcription factor interferon regulatory factor-1 (IRF1) is responsible for IRF8-mediated IL-1 β expression in reactive microglia which are tissue resident macrophages in the central nervous system, although the molecular machineries by which upregulated IRF1 contributes to the expression of IL-1 β remain to be elucidated.²²

[redacted]

Further studies are needed to determine which genes related to IL-1 β secretion are regulated by SGLT2 inhibitor treatment. If the reviewer thinks this RNA seq additional data should be included as a supplementary data, we are willing to do it.

Comment 3: Would empagliflozin in vitro still reduce interleukin-1beta secretion or expression in macrophages, if NLRP3 expression would be silenced beforehand?

Response: Among several inflammasomes, the NLRP3 inflammasome has strong evidence in mouse models and human data that it plays a critical role in the initiation and progress of metabolic disorders²³. Furthermore, the NLRP3 inflammasome is activated in response to the widest array of stimuli such as endogenous damage-associated molecular patterns (DAMPs) (for example, ATP, excess glucose, urate, cholesterol crystals etc.) which are related to various metabolic diseases. Therefore, our aim of the study was to investigate the effect of SGLT2 inhibitor in particular on NLRP3 inflammasome activity (in other words, NLRP3 inflammasome-mediated IL-1 β secretion). Activation of the NLRP3 inflammasome (= secretion of IL-1 β) generally requires two signals: the priming (by LPS in our study) and the triggering (by ATP or palmitate in our study). In previous studies, DAMPs-induced IL-1 β secretion was totally dependent on the NLRP3 (as shown in the figures below).^{24,25}

[redacted]

[redacted]

Therefore, NLRP3-silenced cells cannot induce IL-1 β secretion under signal1/2 stimulation. It is impossible to test whether empagliflozin treatment will further reduce IL-1 β secretion from this condition (NLRP3 silenced cells). Although IL-1 β might be secreted by other activated inflammasomes (e.g. AIM-2) in macrophages, those inflammasome signalings are not our questions to be solved and not closely related to metabolic diseases and ketones.

Comment 4: Although beta-hydroxybutyrate has been described to suppress NLRP3 in mice, what is the mechanisms in human macrophages?

Response: In a paper published in Nature Medicine¹⁷, BHB dose-dependently inhibited IL-1 β secretion not only in mice but also in human monocytes (as a figure below), and which was reproducible in our study (Fig. 4a in the main manuscript).

[redacted]

As mentioned above (response to comment 2), the Nature Medicine's paper showed that BHB blocks NLRP3 inflammasome activation both by controlling an unknown upstream event that reduces K⁺ efflux from macrophages and by inhibiting ASC polymerization, speck formation and assembly of the inflammasome. Although these mechanisms have been elucidated through experiments mainly on mice models, the mechanisms might be same in human macrophages as well, since humans and mice belong to mammals and share similar machinery related to NLRP3 inflammasome.

References

1. Heise, T. *et al.* Safety, tolerability, pharmacokinetics and pharmacodynamics following 4 weeks' treatment with empagliflozin once daily in patients with type 2 diabetes. *Diabetes Obes. Metab.* **15**, 613-621 (2013).
2. Mooradian, A.D., Reed, R.L., Meredith, K.E. & Scuderi, P. Serum levels of tumor necrosis factor and IL-1 alpha and IL-1 beta in diabetic patients. *Diabetes Care* **14**, 63-65 (1991).
3. Doganay, S. *et al.* Comparison of serum NO, TNF-alpha, IL-1beta, sIL-2R, IL-6 and IL-8 levels with grades of retinopathy in patients with diabetes mellitus. *Eye (Lond)* **16**, 163-170 (2002).
4. Gustavsson, C., Agardh, E., Bengtsson, B. & Agardh, C.D. TNF-alpha is an independent serum marker for proliferative retinopathy in type 1 diabetic patients. *J. Diabetes Complications* **22**, 309-316 (2008).
5. Van Tassell, B.W., Toldo, S., Mezzaroma, E. & Abbate, A. Targeting interleukin-1 in heart disease. *Circulation* **128**, 1910-1923 (2013).
6. Wing, M.R. *et al.* Race modifies the association between adiposity and inflammation in patients with chronic kidney disease: findings from the chronic renal insufficiency cohort study. *Obesity (Silver Spring)* **22**, 1359-1366 (2014).
7. Ikonomidis, I. *et al.* Increased proinflammatory cytokines in patients with chronic stable angina and their reduction by aspirin. *Circulation* **100**, 793-798 (1999).
8. Albert, M.A., Danielson, E., Rifai, N. & Ridker, P.M. Effect of statin therapy on C-reactive protein levels: the pravastatin inflammation/CRP evaluation (PRINCE): a randomized trial and cohort study. *JAMA* **286**, 64-70 (2001).
9. Ridker, P.M. *et al.* C-reactive protein levels and outcomes after statin therapy. *N. Engl. J. Med.* **352**, 20-28 (2005).
10. Devlin, C.M., Kuriakose, G., Hirsch, E. & Tabas, I. Genetic alterations of IL-1 receptor antagonist in mice affect plasma cholesterol level and foam cell lesion size. *Proc. Natl. Acad. Sci. U. S. A.* **99**, 6280-6285 (2002).
11. Chamberlain, J. *et al.* Interleukin-1 regulates multiple atherogenic mechanisms in response to fat feeding. *PLoS One* **4**, e5073 (2009).
12. Tan, H.Y. *et al.* Aberrant Inflammasome Activation Characterizes Tuberculosis-Associated Immune Reconstitution Inflammatory Syndrome. *J. Immunol.* **196**, 4052-4063 (2016).
13. Kimmel, P.L., Phillips, T.M., Phillips, E. & Bosch, J.P. Effect of renal replacement therapy on cellular cytokine production in patients with renal disease. *Kidney Int.* **38**,

- 129-135 (1990).
14. Raj, D.S. *et al.* Skeletal muscle, cytokines, and oxidative stress in end-stage renal disease. *Kidney Int.* **68**, 2338-2344 (2005).
 15. Toldo, S. *et al.* Independent roles of the priming and the triggering of the NLRP3 inflammasome in the heart. *Cardiovasc. Res.* **105**, 203-212 (2015).
 16. Di Iorio, A. *et al.* Serum IL-1beta levels in health and disease: a population-based study. 'The InCHIANTI study'. *Cytokine* **22**, 198-205 (2003).
 17. Youm, Y.H. *et al.* The ketone metabolite beta-hydroxybutyrate blocks NLRP3 inflammasome-mediated inflammatory disease. *Nat. Med.* **21**, 263-269 (2015).
 18. Dror, E. *et al.* Postprandial macrophage-derived IL-1beta stimulates insulin, and both synergistically promote glucose disposal and inflammation. *Nat. Immunol.* **18**, 283292 (2017).
 19. DeFronzo, R.A., Norton, L. & Abdul-Ghani, M. Renal, metabolic and cardiovascular considerations of SGLT2 inhibition. *Nat. Rev. Nephrol.* **13**, 11-26 (2017).
 20. Moon, J.S. *et al.* mTORC1-Induced HK1-Dependent Glycolysis Regulates NLRP3 Inflammasome Activation. *Cell Rep.* **12**, 102-115 (2015).
 21. Yoon, B.R. *et al.* Role of SLC7A5 in Metabolic Reprogramming of Human Monocyte/Macrophage Immune Responses. *Front. Immunol.* **9**, 53. (2018).
 22. Tasuda, M. *et al.* Transcription factor IRF1 is responsible for IRF8-mediated IL-1 β expression in reactive microglia. *J Pharmacol Sci.* **128**, 216-220.
 23. Guo, H., Callaway, J.B. & Ting, J.P. Inflammasomes: mechanism of action, role in disease, and therapeutics. *Nat. Med.* **21**, 677-687 (2015).
 24. Karmakar, M. *et al.* Neutrophil P2X7 receptors mediate NLRP3 inflammasome-dependent IL-1 β secretion in response to ATP. *Nat. Commun.* **7**, 10555. (2016).
 25. Duewell, P. *et al.* NLRP3 inflammasomes are required for atherogenesis and activated by cholesterol crystals. *Nature*, **464**, 1357-61 (2010).

REVIEWERS' COMMENTS:

Reviewer #1 (Remarks to the Author):

All my criticisms have been adequately addressed.
Prof David R. Matthews

Reviewer #2 (Remarks to the Author):

Authors have addressed my concerns.

Reviewer #4 (Remarks to the Author):

The authors have been very responsive to the prior critiques and have modified the manuscript appropriately to reflect the limitations of their findings.